# Equivariant flow matching

**Leon Klein**
Freie Universität Berlin
`leon.klein@fu-berlin.de`

**Andreas Krämer**
Freie Universität Berlin
`andreas.kraemer@fu-berlin.de`

**Frank Noé**
Microsoft Research AI4Science
Freie Universität Berlin
Rice University
`franknoe@microsoft.com`

## Abstract

Normalizing flows are a class of deep generative models that are especially interesting for modeling probability distributions in physics, where the exact likelihood of flows allows reweighting to known target energy functions and computing unbiased observables. For instance, Boltzmann generators tackle the long-standing sampling problem in statistical physics by training flows to produce equilibrium samples of many-body systems such as small molecules and proteins. To build effective models for such systems, it is crucial to incorporate the symmetries of the target energy into the model, which can be achieved by equivariant continuous normalizing flows (CNFs). However, CNFs can be computationally expensive to train and generate samples from, which has hampered their scalability and practical application. In this paper, we introduce equivariant flow matching, a new training objective for equivariant CNFs that is based on the recently proposed optimal transport flow matching. Equivariant flow matching exploits the physical symmetries of the target energy for efficient, simulation-free training of equivariant CNFs. We demonstrate the effectiveness of flow matching on rotation and permutation invariant many-particle systems and a small molecule, alanine dipeptide, where for the first time we obtain a Boltzmann generator with significant sampling efficiency without relying on tailored internal coordinate featurization. Our results show that the equivariant flow matching objective yields flows with shorter integration paths, improved sampling efficiency, and higher scalability compared to existing methods.

## 1 Introduction

Generative models have achieved remarkable success across various domains, including images [1, 2, 3, 4], language [5, 6, 7], and applications in the physical sciences [8, 9, 10]. Among the rapidly growing subfields in generative modeling, normalizing flows have garnered significant interest. Normalizing flows [11, 12, 13, 14] are powerful exact-likelihood generative neural networks that transform samples from a simple prior distribution into samples that follow a desired target probability distribution. Previous studies [15, 16, 17, 18, 19] have emphasized the importance of incorporating symmetries of the target system into the flow model. In this work, we focus specifically on many-body systems characterized by configurations $x \in \mathbb{R}^{N \times D}$ of $N$ particles in $D$ spatial dimensions. The symmetries of such systems arise from the invariances of the potential energy function $U(x)$. The

37th Conference on Neural Information Processing Systems (NeurIPS 2023).

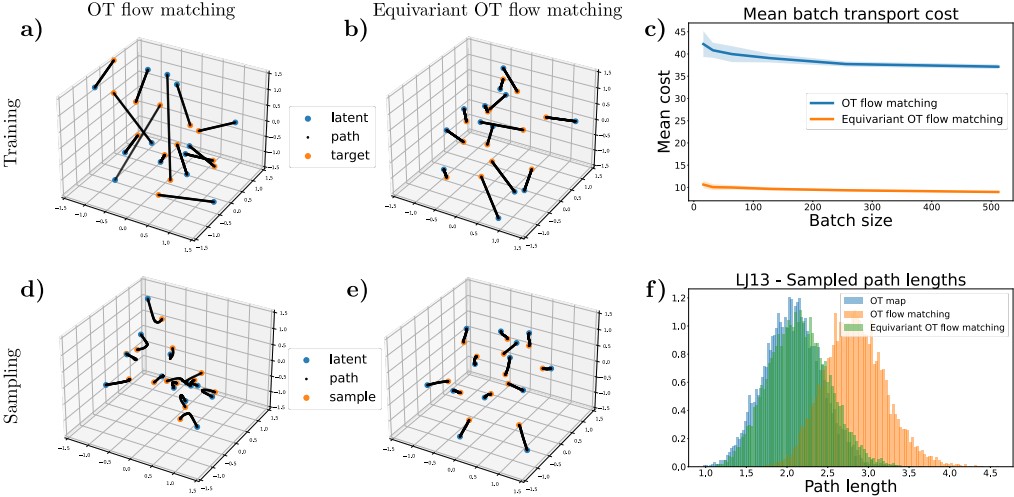

Figure 1: Results for the 13 particle Lennard-Jones system (LJ13) for different flow matching training methods. (a, b) Sample pairs generated with the different flow matching objectives during training. (c) Mean transport cost (squared distance) for training batches. (d, e) Integration paths per particle for samples generated by models trained with OT flow matching and equivariant OT flow matching, respectively. (f) Integration path length, i.e. arc length between prior and output sample, distribution compared with the OT path length between the prior and push-forward distribution of the flows.

associated probability distribution, known as the Boltzmann distribution, is given by

$$\mu(x) \propto \exp\left(-\frac{U(x)}{k_B T}\right), \tag{1}$$

where $k_B$ represents the Boltzmann constant and $T$ denotes the temperature.

Generating equilibrium samples from the Boltzmann distribution is a long-standing problem in statistical physics, typically addressed using iterative methods like Markov Chain Monte Carlo or Molecular Dynamics (MD). In contrast, Boltzmann generators (BGs) [8] utilize normalizing flows to directly generate independent samples from the Boltzmann distribution. Moreover, they allow the reweighting of the generated density to match the unbiased target Boltzmann distribution $\mu$. When dealing with symmetric densities, which are ubiquitous in physical systems, BGs employ two main approaches: (i) describing the system using internal coordinates [8, 20] or (ii) describing the system in Cartesian coordinates while incorporating the symmetries into the flow through equivariant models [17]. In this work, we focus on the latter approach, which appears more promising as such architectures can in principle generalize across different molecules and do not rely on tailored system-specific featurizations. However, current equivariant Boltzmann generators based on continuous normalizing flows (CNFs) face limitations in scalability due to their high computational cost during both training and inference. Recently, flow matching [21] has emerged as a fast, simulation-free method for training CNFs. A most recent extension, optimal transport (OT) flow matching [22], enables learning optimal transport maps, which facilitates fast inference through simple integration paths. In this work, we apply OT flow matching to train flows for highly symmetric densities and find that the required batch size to approximate the OT map adequately can become prohibitively large (Figure 1c). Thus, the learned flow paths will deviate strongly from the OT paths (Figure 1d), which increases computational cost and numerical errors. We tackle this problem by proposing a novel *equivariant* OT flow matching objective for training equivariant flows on invariant densities, resulting in optimal paths for faster inference (see Figure 1e,f).

Our main contributions in this work are as follows:

1. We propose a novel flow matching objective designed for invariant densities, yielding nearly optimal integration paths. Concurrent work of [23] proposes nearly the same objective with the same approximation method, which they also call equivariant flow matching.

2. We compare different flow matching objectives to train equivariant continuous normalizing flows. Through our evaluation, we demonstrate that only our proposed equivariant flow matching objective enables close approximations to the optimal transport paths for invariant densities. However, our proposed method is most effective for larger highly symmetric systems, while achieving sometimes inferior results for the smaller systems compared to optimal transport flow matching.

3. We introduce a new invariant dataset of alanine dipeptide and a large Lennard-Jones cluster. These datasets serve as valuable resources for evaluating the performance of flow models on invariant systems.

4. We present the first Boltzmann Generator capable of producing samples from the equilibrium Boltzmann distribution of a molecule in Cartesian coordinates. Additionally, we demonstrate the reliability of our generator by accurately estimating the free energy difference, in close agreement with the results obtained from umbrella sampling. Concurrent work of [24] also introduce a Boltzmann Generator, based on coupling flows instead of CNFs, in Cartesian coordinates for alanine dipeptide. However, they investigate alanine dipeptide at $T = 800K$ instead of room temperature.

## 2 Related work

Normalizing flows and Boltzmann generators [8] have been applied to molecular sampling and free energy estimation [25, 20, 26, 27, 28]. Previous flows for molecules only achieved significant sampling efficiency when employing either system-specific featurization such as internal coordinates [8, 29, 25, 20, 30, 31] or prior distributions close to the target distribution [32, 31, 33]. Notably, our equivariant OT flow matching method could also be applied in such scenarios, where the prior distribution is sampled by MD at a different thermodynamic state (e.g., higher temperature or lower level of theory). A flow model for molecules in Cartesian coordinates has been developed in [34] where a transferable coupling flow is used to sample small peptide conformations by proposing iteratively large time steps instead of sampling from the target distribution directly. Equivariant diffusion models [35, 36] learn a score-based model to generate molecular conformations. The score-based model is parameterized by an equivariant function similar to the vector field used in equivariant CNFs. However, they do not target the Boltzmann distribution. As an exception, [37] propose a diffusion model in torsion space and use the underlying probability flow ODE as a Boltzmann generator. Moreover, [38] use score-based models to learn the transition probability for multiple time-resolutions, accurately capturing the dynamics.

To help speed up CNF training and inference, various authors [39, 40, 41] have proposed incorporating regularization terms into the likelihood training objective for CNFs to learn an optimal transport (OT) map. While this approach can yield flows similar to OT flow matching, the training process itself is computationally expensive, posing limitations on its scalability. The general idea underlying flow matching was independently conceived by different groups [21, 42, 43] and soon extended to incorporate OT [22, 44]. OT maps with invariances have been studied previously to map between learned representations [45], but have not yet been applied to generative models. Finally, we clarify that we use the term flow matching exclusively to refer to the training method for flows introduced by Lipman et al. [21] rather than the flow-matching method for training energy-based coarse-grained models from flows introduced at the same time [46].

## 3 Method

In this section, we describe the key methodologies used in our study, highlighting important prior work.

### 3.1 Normalizing flows

Normalizing flows [14, 47] provide a powerful framework for learning complex probability densities $\mu(x)$ by leveraging the concept of invertible transformations. These transformations, denoted as $f_\theta : \mathbb{R}^n \to \mathbb{R}^n$, map samples from a simple prior distribution $q(x_0) = \mathcal{N}(x_0|0, I)$ to samples from a more complicated output distribution. This resulting distribution $p(x_1)$, known as the *push-forward*

distribution, is obtained using the *change of variable* equation:

$$p(x_1) = q\left(f_\theta^{-1}(x_1)\right)\det\left|\frac{\partial f_\theta^{-1}(x_1)}{\partial x_1}\right|, \tag{2}$$

where $\frac{\partial f_\theta^{-1}(x_1)}{\partial x_1}$ denotes the Jacobian of the inverse transformation $f_\theta^{-1}$.

## 3.2 Continuous normalizing flows (CNFs)

CNFs [48, 49] are an instance of normalizing flows that are defined via the ordinary differential equation

$$\frac{df_\theta^t(x)}{dt} = v_\theta\left(t, f_\theta^t(x)\right), \quad f_\theta^0(x) = x, \tag{3}$$

where $v_\theta(t, x) : \mathbb{R}^n \times [0, 1] \to \mathbb{R}^n$ is a time dependent vector field. The time dependent flow $f_\theta^t(x)$ generates the probability flow path $\tilde{p}_t(x) : [0, 1] \times \mathbb{R}^n \to \mathbb{R}^+$ from the prior distribution $\tilde{p}_0(x) = q(x_0)$ to the push-forward distribution $\tilde{p}_1(x)$. We can generate samples from $\tilde{p}_t(x)$ by first sampling $x_0$ from the prior and then integrating

$$f_\theta^t(x) = x_0 + \int_0^t dt' v_\theta\left(t', f_\theta^{t'}(x)\right). \tag{4}$$

The corresponding log density change is given by the continuous change of variable equation

$$\log \tilde{p}_t(x) = \log q(x_0) - \int_0^t dt' \nabla \cdot v_\theta\left(t', f_\theta^{t'}(x)\right), \tag{5}$$

where $\nabla \cdot v_\theta\left(t', f_\theta^{t'}(x)\right)$ is the trace of the Jacobian.

## 3.3 Likelihood-based training

CNFs can be trained using a likelihood-based approach. In this training method, the loss function is formulated as the negative log-likelihood

$$\mathcal{L}_{\text{NLL}}(\theta) = \mathbb{E}_{x_1 \sim \mu}\left[-\log q\left(f_\theta^{-1}(x_1)\right) - \log\left|\det \frac{\partial f_\theta^{-1}(x_1)}{\partial x_1}\right|\right]. \tag{6}$$

Evaluating the inverse map requires propagating the samples through the reverse ODE, which involves numerous evaluations of the vector field for integration in each batch. Additionally, evaluating the change in log probability involves computing the expensive trace of the Jacobian.

## 3.4 Flow matching

In contrast to likelihood-based training, flow matching allows training CNFs simulation-free, i.e. without integrating the vector field and evaluating the Jacobian, making the training significantly faster (see Appendix A.2). Hence, the flow matching enables scaling to larger models and larger systems with the same computational budget. The flow matching training objective [21, 22] regresses $v_\theta(t, x)$ to some target vector field $u_t(x)$ via

$$\mathcal{L}_{\text{FM}}(\theta) = \mathbb{E}_{t \sim [0,1], x \sim p_t(x)} ||v_\theta(t, x) - u_t(x)||_2^2. \tag{7}$$

In practice, $p_t(x)$ and $u_t(x)$ are intractable, but it has been shown in [21] that the same gradients can be obtained by using the conditional flow matching loss:

$$\mathcal{L}_{\text{CFM}}(\theta) = \mathbb{E}_{t \sim [0,1], x \sim p_t(x|z)} ||v_\theta(t, x) - u_t(x|z)||_2^2. \tag{8}$$

The vector field $u_t(x)$ and corresponding probability path $p_t(x)$ are given in terms of the conditional vector field $u_t(x|z)$ as well as the conditional probability path $p_t(x|z)$ as

$$p_t(x) = \int p_t(x|z)p(z)dz \quad \text{and} \quad u_t(x) = \int \frac{p_t(x|z)u_t(x|z)}{p_t(x)}p(z)dz, \tag{9}$$

where $p(z)$ is some arbitrary conditioning distribution independent of $x$ and $t$. For a derivation see [21] and [22]. There are different ways to efficiently parametrize $u_t(x|z)$ and $p_t(x|z)$. We here focus on a parametrization that gives rise to the optimal transport path, as introduced in [22]

$$z = (x_0, x_1) \quad \text{and} \quad p(z) = \pi(x_0, x_1) \tag{10}$$

$$u_t(x|z) = x_1 - x_0 \quad \text{and} \quad p_t(x|z) = \mathcal{N}(x|t \cdot x_1 + (1 - t) \cdot x_0, \sigma^2), \tag{11}$$

where the conditioning distribution is given by the 2-Wasserstein optimal transport map $\pi(x_0, x_1)$ between the prior $q(x_0)$ and the target $\mu(x_1)$. The 2-Wasserstein optimal transport map is defined by the 2-Wasserstein distance

$$W_2^2 = \inf_{\pi \in \Pi} \int c(x_0, x_1) \pi(dx_0, dx_1), \tag{12}$$

where $\Pi$ is the set of couplings as usual and $c(x_0, x_1) = ||x_0 - x_1||_2^2$ is the squared Euclidean distance in our case. Following [22], we approximate $\pi(x_0, x_1)$ by only considering a batch of prior and target samples. Hence, for each batch we generate samples from $p(z)$ as follows:

1. sample batches of points $(x_0^1, \ldots, x_0^B) \sim q$ and $(x_1^1, \ldots, x_1^B) \sim \mu$,
2. compute the cost matrix $M$ for the batch, i.e. $M_{ij} = ||x_0^i - x_1^j||_2^2$,
3. solve the discrete OT problem defined by $M$,
4. generate training pairs $z^i = (x_0^i, x_1^i)$ according to the OT solution.

We will refer to this training procedure as *OT flow matching*.

### 3.5 Equivariant flows

Symmetries can be described in terms of a *group $G$* acting on a finite-dimensional vector space $V$ via a matrix representation $\rho(g); g \in G$. A map $I : V \to V'$ is called $G$-invariant if $I(\rho(g)x) = I(x)$ for all $g \in G$ and $x \in V$. Similarly, a map $f : V \to V$ is called $G$-equivariant if $f(\rho(g)x) = \rho(g)f(x)$ for all $g \in G$ and $x \in V$.

In this work, we focus on systems with energies $U(x)$ that exhibit invariance under the following symmetries: (i) *Permutations* of interchangeable particles, described by the symmetric group $S(N')$ for each interchangeable particle group. (ii) *Global rotations and reflections*, described by the orthogonal group $O(D)$ (iii) *Global translations*, described by the translation group $\mathbb{T}$. These symmetries are commonly observed in many particle systems, such as molecules or materials.

In [17], it is shown that equivariant CNFs can be constructed using an equivariant vector field $v_\theta$ (Theorem 2). Moreover, [17, 50] show that the push-forward distribution $\tilde{p}(x_1)$ of a $G$-equivariant flow with a $G$-invariant prior density is also $G$-invariant (Theorem 1). Note that this is only valid for orthogonal maps, and hence not for translations. However, translation invariance can easily be achieved by assuming mean-free systems as proposed in [17]. An additional advantage of mean-free systems is that global rotations are constrained to occur around the origin. By ensuring that the flow does not modify the geometric center and the prior distribution is mean-free, the resulting push-forward distribution will also be mean-free.

Although equivariant flows can be successfully trained with the OT flow matching objective, the trained flows display highly curved vector fields (Figure 1d) that do not match the linear OT paths. Fortunately, the OT training objective can be modified so that it yields better approximations to the OT solution for finite batch sizes.

## 4 Equivariant optimal transport flow matching

Prior studies [51, 22] indicate that medium to small batch sizes are often sufficient for effective OT flow matching. However, when dealing with highly symmetric densities, accurately approximating the OT map may require a prohibitively large batch size. This is particularly evident in cases involving permutations, where even for small system sizes, it is unlikely for any pair of target-prior samples $(x_0, x_1)$ to share the exact same permutation. This is because the number of possible permutations scales with the number of interchangeable particles as $N!$, while the number of combinations of the sample pairs scales only with squared batch size (Figure 1). To address this challenge, we propose using a cost function

$$\tilde{c}(x_0, x_1) = \min_{g \in G} ||x_0 - \rho(g)x_1||_2^2, \tag{13}$$

that accurately accounts for the underlying symmetries of the problem in the OT flow matching algorithm. Hence, instead of solely reordering the batch, we instead also align samples along their orbits.

We summarize or main theoretical findings in the following theorem.

**Theorem 1.** *Let $G$ be a compact group that acts on an Euclidean $n$-space by isometries. Let $T\colon x \mapsto y$ be an OT map between $G$-invariant measures $\nu_1$ and $\nu_2$, using the cost function $c$. Then*

1. *$T$ is $G$-equivariant and the corresponding OT plan $\pi(\nu_1, \nu_2)$ is $G$-invariant.*

2. *For all pairs $(x, T(x))$ and $y \in G \cdot T(x)$ :*

$$c(x, T(x)) = \int_G c(g \cdot x, g \cdot T(x)) d\mu(g) = \min_{g \in G} c(x, g \cdot y) \tag{14}$$

3. *$T$ is also an OT map for the cost function $\tilde{c}$.*

Refer to Appendix B.1 for an extensive derivation and discussion. The key insights can be summarized as follows: (i) Given the $G$-equivariance of the target OT map $T$, it is natural to employ an $G$-equivariant flow model for its learning. (ii) From 2. and 3., we can follow that our proposed cost function $\tilde{c}$, effectively aligns pairs of samples in a manner consistent with how they are aligned under the $G$-equivariant OT map $T$.

In this work we focus on $O(D)$- and $S(N)$-invariant distributions, which are common for molecules and multi-particle systems. The minimal squared distance for a pair of points $(x_0, x_1)$, taking into account these symmetries, can be obtained by minimizing the squared Euclidean distance over all possible combinations of rotations, reflections, and permutations

$$\tilde{c}(x_0, x_1) = \min_{r \in O(D), s \in S(N)} ||x_0 - \rho(rs)x_1||_2^2. \tag{15}$$

However, computing the exact minimal squared distance is computationally infeasible in practice due to the need to search over all possible combinations of rotations and permutations. Therefore, we approximate the minimizer by performing a sequential search

$$\tilde{c}(x_0, x_1) = \min_{r \in SO(D)} ||x_0 - \rho(r\tilde{s})x_1||_2^2, \quad \tilde{s} = \arg\min_{s \in S(N)} ||x_0 - \rho(s)x_1||_2^2. \tag{16}$$

We demonstrate in Section 6 that this approximation results in nearly OT integration paths for equivariant flows, even for small batch sizes. While we also tested other approximation strategies in Appendix A.9, they did not yield significant changes in our results, but come at with additional computational overhead. We hence alter the OT flow matching procedure as follows: For each element of the cost matrix $M$, we first compute the optimal permutation with the Hungarian algorithm [52] and then align the two samples through rotation with the Kabsch algorithm [53]. The other steps of the OT flow matching algorithm remain unchanged. We will refer to this loss as *Equivariant OT flow matching*. Although aligning samples in that way comes with a significant computational overhead, this reordering can be done in parallel before or during training (see Appendix C.4). It is worth noting that in the limit of infinite batch sizes, the Euclidean cost will still yield the correct OT map for invariant densities.

## 5 Architecture

The vector field $v_\theta(t, x)$ is parametrized by an $O(D)$- and $S(N)$-equivariant graph neural network, similar to the one used in [18] and introduced in [54]. The graph neural network consists of $L$ consecutive layers. The update for the $i$-th particle is computed as follows

$$h_i^0 = (t, a_i), \quad m_{ij}^l = \phi_e\left(h_i^l, h_j^l, d_{ij}^2\right), \tag{17}$$

$$x_i^{l+1} = x_i^l + \sum_{j \neq i} \frac{(x_i^l - x_j^l)}{d_{ij} + 1} \phi_d(m_{ij}^l), \tag{18}$$

$$h_i^{l+1} = \phi_h\left(h_i^l, m_i^l\right), \quad m_i^l = \sum_{j \neq i} \phi_m(m_{ij}^l) m_{ij}^l, \tag{19}$$

$$v_\theta(t, x^0)_i = x_i^L - x_i^0, \tag{20}$$

Table 1: Comparison of flows trained with different training objectives. Errors are computed over three runs.

| Training type | NLL ($\downarrow$) | ESS ($\uparrow$) | Path length ($\downarrow$) |
|---|---|---|---|
| | DW4 | | |
| Likelihood [18] | $1.72 \pm 0.01$ | $86.87 \pm 0.19\%$ | $3.11 \pm 0.04$ |
| OT flow matching | $1.70 \pm 0.02$ | $\mathbf{92.37 \pm 0.89}\%$ | $2.94 \pm 0.02$ |
| Equivariant OT flow matching | $1.68 \pm 0.01$ | $88.71 \pm 0.40\%$ | $2.92 \pm 0.01$ |
| | LJ13 | | |
| Likelihood [18] | $-15.83 \pm 0.07$ | $39.78 \pm 6.19\%$ | $5.08 \pm 0.22$ |
| OT flow matching | $-16.09 \pm 0.03$ | $54.36 \pm 5.43\%$ | $2.84 \pm 0.01$ |
| Equivariant OT flow matching | $-16.07 \pm 0.02$ | $\mathbf{57.98 \pm 2.19}\%$ | $\mathbf{2.15 \pm 0.01}$ |
| | LJ55 | | |
| OT flow matching | $-88.45 \pm 0.04$ | $3.74 \pm 1.06\%$ | $7.53 \pm 0.02$ |
| Equivariant OT flow matching | $\mathbf{-89.27 \pm 0.04}$ | $\mathbf{4.42 \pm 0.35}\%$ | $\mathbf{3.52 \pm 0.01}$ |
| | Alanine dipeptide | | |
| OT flow matching | $\mathbf{-107.54 \pm 0.03}$ | $\mathbf{1.41 \pm 0.04}\%$ | $10.19 \pm 0.03$ |
| Equivariant OT flow matching | $-106.78 \pm 0.02$ | $0.69 \pm 0.05\%$ | $\mathbf{9.46 \pm 0.01}$ |

where $\phi_\alpha$ are neural networks, $d_{ij}$ is the Euclidean distance between particle $i$ and $j$, and $a_i$ is an embedding for the particle type. Notably, the update conserves the geometric center if all particles are of the same type (see Appendix B.2), otherwise we subtract the geometric center after the last layer. This ensures that the resulting equivariant vector field $v_\theta(t, x)$ conserves the geometric center. When combined with a symmetric mean-free prior distribution, the push-forward distribution of the CNF will be $O(D)$- and $S(N)$-invariant.

## 6 Experiments

In this section, we demonstrate the advantages of equivariant OT flow matching over existing training methods using four datasets characterized by invariant energies. We explore three different training objectives: (i) likelihood-based training, (ii) OT flow matching, and (iii) equivariant OT flow matching. For a comprehensive overview of experimental details, including dataset parameters, error bars, learning rate schedules, computing infrastructure, and additional experiments, please refer to Appendix A and Appendix C in the supplementary material. For all experiments, we employ a mean-free Gaussian prior distribution. The number of layers and parameters in the equivariant CNF vary across datasets while remaining consistent within each dataset (see Appendix C). We provide naïve flow matching as an additional baseline in Appendix A.11. However, the results are very similar to the ones obtained with OT flow matching, although the integration paths are generally slightly longer.

### 6.1 DW4 and LJ13

We first evaluate the different loss functions on two many-particle systems, DW4 and LJ13, that were specifically designed for benchmarking equivariant flows as described in [17]. These systems feature pair-wise double-well and Lennard-Jones interactions with 4 and 13 particles, respectively (see Appendix C.2 for more details). While state-of-the-art results have been reported in [18], their evaluations were performed on very small test sets and were biased for the LJ13 system. To provide a fair comparison, we retrain their model using likelihood-based training as well as the two flow matching losses on resampled training and test sets for both systems. Additionally, we demonstrate improved likelihood performance of flow matching on their biased test set in Appendix A.3. The results, presented in Table 1, show that the two flow matching objectives outperform likelihood-based training while being computationally more efficient. The effective sample sizes (ESS) and negative log likelihood (NLL) are comparable for the flow matching runs. However, for the LJ13 system, the equivariant OT flow matching objective significantly reduces the integration path length compared

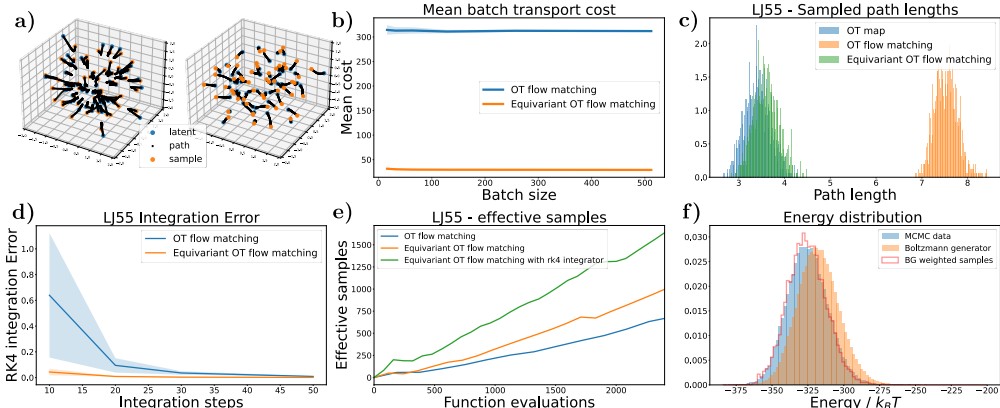

Figure 2: Results for the LJ55 system (a) Integration paths per particle for OT flow matching (left) and equivariant OT flow matching (right). (b) Mean transport cost (squared distance) for training batches. (c) Integration path length distribution. (d) Integration error for a fixed step size integrator (rk4) with respect to a reference solution generated by an adaptive solver (dropi5). (e) Effective samples vs number of function evaluations, i.e. evaluations of the vector field, for a sampling batch size of 1000. (f) Energy histograms for a flow trained with equivariant OT flow matching.

to other methods due to the large number of possible permutations (refer to Figure 1 for a visual illustration).

## 6.2 LJ55

The effectiveness of equivariant OT flow matching becomes even more pronounced when training on larger systems, where likelihood training is infeasible (see Appendix A.2). To this end, we investigate a large Lennard-Jones cluster with 55 particles (*LJ55*).

We observe that the mean batch transport cost of training pairs is about 10 times larger for OT flow matching compared to equivariant OT flow matching (Figure 2b), resulting in curved and twice as long integration paths during inference (Figure 2a,c). However, the integration paths for OT flow matching are shorter than those seen during training, see Appendix A.1 for a more detailed discussion. We compare the integration error caused by using a fixed step integrator (rk4) instead of an adaptive solver (dropi5 [55]). As the integration paths follow nearly straight lines for the equivariant OT flow matching, the so resulting integration error is minimal, while the error is significantly larger for OT flow matching (Figure 2d). Hence, we can use a fixed step integrator, with e.g. 20 steps, for sampling for equivariant OT flow matching, resulting in a three times speed-up over OT flow matching (Figure 2e), emphasizing the importance of accounting for symmetries in the loss for large, highly symmetric systems. Moreover, the equivariant OT flow matching objective outperforms OT flow matching on all evaluation metrics (Table 1). To ensure that the flow samples all states, we compute the energy distribution (Figure 2f) and perform deterministic structure minimization of the samples, similar to [17], in Appendix A.7.

## 6.3 Alanine dipeptide

In our final experiment, we focus on the small molecule alanine dipeptide (Figure 3a) in Cartesian coordinates. The objective is to train a Boltzmann Generator capable of sampling from the equilibrium Boltzmann distribution defined by the semi-empirical *GFN2-xTB* force-field [56]. This semi-empirical potential energy is invariant under permutations of atoms of the same chemical element and global rotations and reflections. However, here we consider the five backbone atoms defining the $\varphi$ and $\psi$ dihedral angles as distinguishable to facilitate analysis. Since running MD simulations with semi-empirical force-fields is computationally expensive, we employ a surrogate training set, further challenging the learning task.

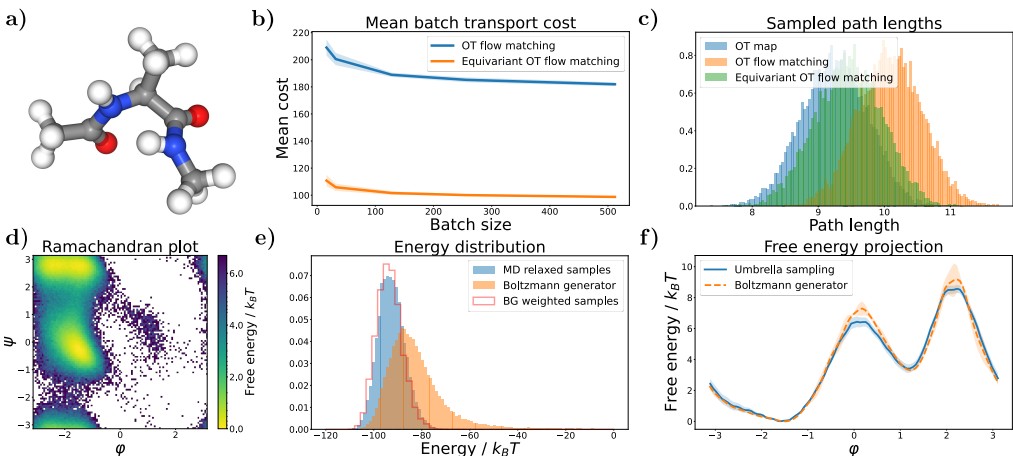

Figure 3: Results for the alanine dipeptide system (a) Alanine dipeptide molecule. (b) Mean transport cost (squared distance) for training batches. (c) Integration path length distribution. (d) Ramachandran plot depicting the generated joint marginal distribution over the backbone dihedral angles $\varphi$ and $\psi$ after filtering out samples with right-handed chirality and high energies. (e) Energy histograms for samples generated by a flow trained with OT flow matching. (f) Free energy distribution along the slowest transition ($\varphi$ dihedral angle) computed with umbrella sampling and the equivariant flow.

**Alanine dipeptide data set generation**    The alanine dipeptide training data set is generated through two steps: (i) Firstly, we perform an MD simulation using the classical *Amber ff99SBildn* force-field for a duration of 1 ms [20]. (ii) Secondly, we relax $10^5$ randomly selected states from the MD simulation using the semi-empirical *GFN2-xTB* force-field for 100 fs each. For more detailed information, refer to Appendix C.2 in the supplementary material. This training data generation is significantly cheaper than performing a long MD simulation with the semi-empirical force field.

Although the Boltzmann generator is trained on a biased training set, we can generate asymptotically unbiased samples from the semi-empirical target distribution by employing reweighting (Appendix B.3) as demonstrated in Figure 3e. In this experiment, OT flow matching outperforms equivariant OT flow matching in terms of effective sample size (ESS) and negative log likelihood (NLL) as shown in Table 1. However, the integration path lengths are still longer for OT flow matching compared to equivariant OT flow matching, as depicted in Figure 3c. Since the energy of alanine dipeptide is invariant under global reflections, the flow generates samples for both chiral states. While the presence of the mirror state reflects the symmetries of the energy, in practical scenarios, molecules typically do not change their chirality spontaneously. Therefore, it may be undesirable to have samples from both chiral states. However, it is straightforward to identify samples of the undesired chirality and apply a mirroring operation to correct them. Alternatively, one may use a SO(3) equivariant flow without reflection equivariance [54] to prevent the generation of mirror states.

**Alanine dipeptide - free energy difference**    The computation of free energy differences is a common challenge in statistical physics as it determines the relative stability of metastable states. In the specific case of alanine dipeptide, the transition between negative and positive $\varphi$ dihedral angle is the slowest process (Figure 3d), and equilibrating the free energy difference between these two states from MD simulation requires simulating numerous transitions, i.e. millions of consecutive MD steps, which is expensive for the semi-empirical force field. In contrast, we can train a Boltzmann generator from data that is not in global equilibrium, i.e. using our biased training data, which is significantly cheaper to generate. Moreover, as the positive $\varphi$ state is much less likely, we can even bias our training data to have nearly equal density in both states, which helps compute a more precise free energy estimate (see Appendix C.5). The equivariant Boltzmann generator is trained using the OT flow matching loss, which exhibited slightly better performance for alanine dipeptide. We obtain similar results for the equivariant OT flow matching loss (see Appendix A.4). To obtain an accurate estimation of the free energy difference, five umbrella sampling simulations are conducted along the $\varphi$ dihedral angle using the semi-empirical force-field (see Appendix C.2). The free energy difference estimated by the Boltzmann generator demonstrates good agreement with the results

Table 2: Dimensionless free energy differences for the slowest transition of alanine dipeptide estimated from various methods. Umbrella sampling yields a converged reference solution. Errors over five runs.

|  | MD | relaxed MD | Umbrella sampling | Boltzmann generator |
| --- | --- | --- | --- | --- |
| Free energy difference / $k_B T$ | 5.31 | 5.00 | $4.10 \pm 0.26$ | $4.10 \pm 0.08$ |

of these simulations (Table 2 and Figure 3f), whereas both the relaxed training data and classical Molecular Dynamics simulation overestimate the free energy difference.

## 7 Discussion

We have introduced a novel flow matching objective for training equivariant continuous normalizing flows on invariant densities, leading to optimal transport integration paths even for small training batch sizes. By leveraging flow matching objectives, we successfully extended the applicability of equivariant flows to significantly larger systems, including the large Lennard-Jones cluster (LJ55). We conducted experiments comparing different training objectives on four symmetric datasets, and our results demonstrate that as the system size increases, the importance of accounting for symmetries within the flow matching objective becomes more pronounced. This highlights the critical role of our proposed flow matching objective in scaling equivariant CNFs to even larger systems.

Another notable contribution of this work is the first successful application of a Boltzmann generator to model alanine dipeptide in Cartesian coordinates. By accurately estimating free energy differences using a semi-empirical force-field, our approach of applying OT flow matching and equivariant OT flow matching to equivariant flows demonstrates its potential for reliable simulations of complex molecular systems.

## 8 Limitations / Future work

While we did not conduct experiments to demonstrate the transferability of our approach, the architecture and proposed loss function can potentially be used to train transferable models. We leave this for future research. Although training with flow matching is faster and computationally cheaper than likelihood training for CNFs, the inference process still requires the complete integration of the vector field, which can be computationally expensive. However, if the model is trained with the equivariant OT flow matching objective, faster fixed-step integrators can be employed during inference. Our suggested approximation of Equation (15) includes the Hungarian algorithm, which has a computational complexity of $\mathcal{O}(N^3)$. To improve efficiency, this step could be replaced by heuristics relying on approximations to the Hungarian algorithm [57]. Moreover, flow matching does not allow for energy based training, as this requires integration similar to NLL training. A potential alternative approach is to initially train a CNF using flow matching with a small set of samples. Subsequent sample generation through the CNF, followed by reweighting to the target distribution, allows these samples to be added iteratively to the training set, similarly as in [27, 37]. Notably, in our experiments, we exclusively examined Gaussian prior distributions. However, flow matching allows to transform arbitrary distributions. Therefore, our equivariant OT flow matching method holds promise for application in scenarios where the prior distribution is sampled through MD at a distinct thermodynamic state, such as a higher temperature or a different level of theory [32, 33]. In these cases, where both the prior and target distributions are close, with samples alignable through rotations and permutations, we expect that the advantages of equivariant OT flow matching will become even more pronounced than what we observed in our experiments.

Building upon the success of training equivariant CNFs for larger systems using our flow matching objective, future work should explore different architectures for the vector field. Promising candidates, such as [58, 59, 60, 61, 62], could be investigated to improve the modeling capabilities of equivariant CNFs. While our focus in this work has been on symmetric physical systems, it is worth noting that equivariant flows have applications in other domains that also exhibit symmetries, such as traffic data generation [63], point cloud and set modeling [64, 65], as well as invariant distributions on arbitrary manifolds [66]. Our equivariant OT flow matching approach can be readily applied to these areas of application without modification.

## Acknowledgements

We gratefully acknowledge support by the Deutsche Forschungsgemeinschaft (SFB1114, Projects No. C03, No. A04, and No. B08), the European Research Council (ERC CoG 772230 "ScaleCell"), the Berlin Mathematics center MATH+ (AA1-6), and the German Ministry for Education and Research (BIFOLD - Berlin Institute for the Foundations of Learning and Data). We gratefully acknowledge the computing time granted by the Resource Allocation Board and provided on the supercomputer Lise at NHR@ZIB and NHR@Göttingen as part of the NHR infrastructure. We thank Maaike Galama, Michele Invernizzi, Jonas Köhler, and Max Schebek for insightful discussions. Moreover, we would like to thank Lea Zimmermann and Felix Draxler, who tested our code and brought to our attention a critical bug.

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

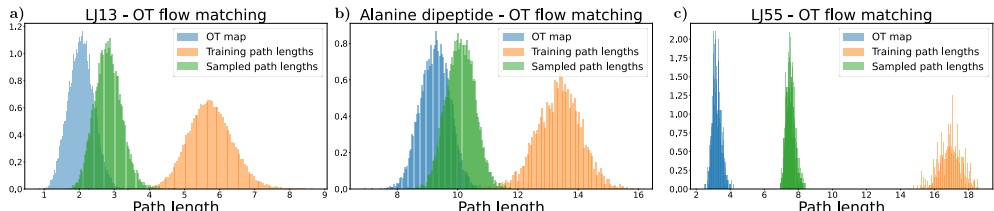

Figure 4: Path lengths distributions for models trained with OT flow matching. a) LJ13 system b) Alanine dipeptide system c) LJ55 system.

# Supplementary Material

## A    Additional experiments

### A.1    Equivariant flows can achieve better paths than they are trained for

We make an interesting observation regarding the performance of equivariant flows trained with the OT flow matching objective. As shown in Figure 4a-c, these flows generate integration paths that are significantly shorter compared to their training counterparts. While they do not learn the OT path and exhibit curved trajectories with notable directional changes (as seen in Figure 1d and Figure 2a), an interesting finding is the absence of path crossings. Any permutation of the samples would increase the distance between the latent sample and the push-forward sample. To quantify the length of these paths, we compute the Euclidean distance between the latent sample and the push-forward sample. The relatively short path lengths can be attributed, in part, to the architecture of the flow model. As the flow is equivariant with respect to permutations, it prevents the crossing of particle paths and forces the particles to change the direction instead, as an average vector field is learned.

### A.2    Memory requirement

As the system size increases, the memory requirements for likelihood training become prohibitively large. In the case of the LJ55 system, even a batch size of 5 necessitates more than 24 GB of memory, making effective training infeasible. In contrast, utilizing flow matching enables training the same model with a batch size of 256 while requiring less than 12 GB of memory.

For the alanine dipeptide system, likelihood training with a batch size of 8 demands more than 12 GB of memory, and a batch size of 16 exceeds 24 GB. On the other hand, employing flow matching allows training the same model with a batch size of 256, consuming less than 3 GB of memory.

The same holds for energy based training, as we need to evaluate Equation (3) and Equation (4) as well during training. Therefore, we opt to solely utilize flow matching for these two datasets.

### A.3    Comparison with prior work on equivariant flows

Both [17] and [18] evaluate their models on the DW4 and LJ13 systems. However, [17] do not provide the numerical values for the negative log likelihood (NLL), only presenting them visually. On the other hand, [18] use a small test set for both experiments. Additionally, their test set for LJ13 is biased as it originates from the burn-in phase of the sampling.

Although the test sets for both systems are insufficient, we still compare our OT flow matching and equivariant OT flow matching to these previous works in Table 3. It should be noted that the value range differs significantly from the values reported in Table 1, as the likelihood in this case includes the normalization constant of the prior distribution.

### A.4    Alanine dipeptide - free energy difference

Here we present additional results for the free energy computation of alanine dipeptide, as discussed in Section 6.3. The effective sample size for the trained models is $0.50 \pm 0.13\%$, which is lower due to training on the biased dataset (compare to Table 1). We illustrate the Ramachandran plot in

Table 3: Comparison of different training methods for the DW4 and (biased) LJ13 dataset as used in [18] for $10^5$ and $10^4$ training points, respectively.

| Dataset | Likelihood [18] | OT flow matching | Equivariant OT flow matching |
|---|---|---|---|
| DW4 NLL ($\downarrow$) | $7.48 \pm 0.05$ | $7.21 \pm 0.03$ | $7.18 \pm 0.02$ |
| LJ13 NLL ($\downarrow$) | $30.41 \pm 0.16$ | $30.34 \pm 0.09$ | $30.30 \pm 0.03$ |

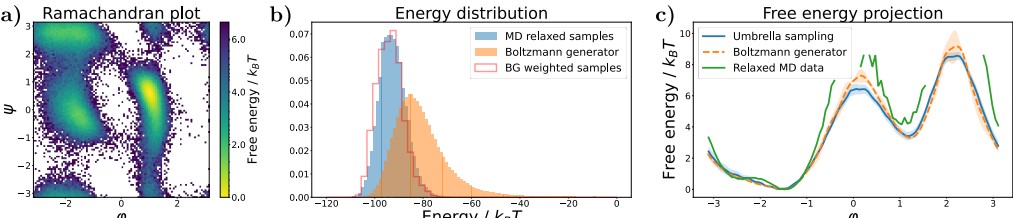

Figure 5: Alanine dipeptide - free energy experiments. The model is trained on the biased dataset. (a) Ramachandran plot depicting the generated joint marginal distribution over the backbone dihedral angles $\varphi$ and $\psi$ after filtering out samples with right-handed chirality and high energies. (b) Energy histograms for samples generated by a flow trained with OT flow matching. (c) Free energy distribution along the slowest transition ($\varphi$ dihedral angle) computed with umbrella sampling, the equivariant flow, and a relaxed MD trajectory.

Figure 5a, the energy distribution in Figure 5b, and the free energy projections for Umbrella sampling, model samples, and relaxed MD samples in Figure 5c.

We obtain similar results with equivariant OT flow matching, as shown in Table 4.

### A.5 Alanine dipeptide integration paths

We compare the integration paths of the models trained on the alanine dipeptide dataset in Figure 6a,b. Despite the higher number of atoms in the alanine dipeptide system (22 atoms), we observe a comparable difference in the integration paths as observed in the LJ13 system. This similarity arises from the presence of different particle types in the alanine dipeptide, resulting in a similar number of possible permutations compared to the LJ13 system.

### A.6 Integration error

The integration error, which arises from the use of a fixed-step integration method (rk4) instead of an adaptive solver (dopri5), is quantified by calculating the sum of absolute differences in sample positions and changes in log probability (dlogp). The discrepancies in position are depicted in Figure 2d, while the difference in log probability change are presented in Figure 6c. Both of these are crucial in ensuring the generation of unbiased samples that accurately represent the target Boltzmann distribution of interest.

### A.7 Structure minimization

Following a similar approach as the deterministic structure minimization conducted for the LJ13 system in [17], we extend our investigation to the sampled minima of the LJ55 system. We minimize the energy of samples generated by a model trained with equivariant OT flow matching, as well as samples obtained from the test data. The resulting structures represent numerous local minima within the potential energy distribution.

In Figure 6d, we observe a good agreement between the distributions obtained from both methods. Furthermore, it is noteworthy that both approaches yield samples corresponding to the global minimum. The energies of the samples at 300K are considerably higher than those of the minimized structures.

Table 4: Dimensionless free energy differences for the slowest transition of alanine dipeptide estimated with a Boltzmann Generator trained with the equivariant OT flow matching objective. Umbrella sampling yields a converged reference solution. Errors over five runs.

|  | Umbrella sampling | Equivariant OT flow matching |
|---|---|---|
| Free energy difference / $k_B T$ | $4.10 \pm 0.26$ | $3.95 \pm 0.19$ |

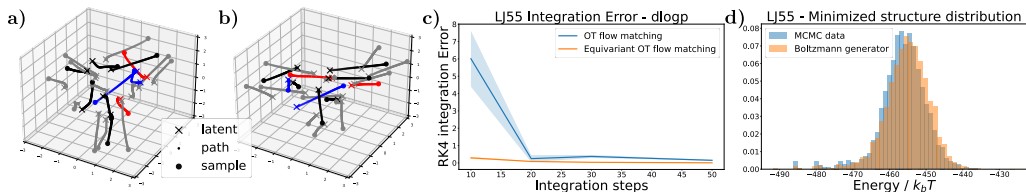

Figure 6: (a,b) Integration paths for alanine dipeptide for models trained with OT flow matching and equivariant OT flow matching, respectively. The color code for the atoms is as usual: gray - H, black - C, blue - N, red - O. (c) Integration error for a fixed step integrator for the LJ55 system. (d) Minimized samples from a model trained on the LJ55 system with equivariant OT flow matching.

## A.8 Log weight distribution

Samples generated with a trained Boltzmann generators can be reweighted to the target distribution via Equation (29). Generally, the lower the variance of the log weight distribution, the higher the ESS. Although large outliers might reduce the ESS significantly. We present the log weight distribution for the LJ55 system in Figure 7a, showing that the weight distribution for the BG trained with equivariant OT flow matching has lower variance.

## A.9 Approximation methods for equivariant OT flow matching

The approximation given in Equation (16) is in practice performed using the Hungarian algorithm for permutations and the Kabsch algorithm for rotations. However, one could also think of other approximations, such as performing the rotation first or alternating multiple permutations and rotations. We compare different approximation strategies in Figure 7b,c. The baseline reference is computed with an expensive search over the approximation given in Equation (16). Namely, we evaluate Equation (16) for 100 random rotations combined with the global reflection, denoted as $O_{200}$, for each sample, i.e.

$$\hat{c}(x_0, x_1) = \min_{o \in O_{200}(D)} \tilde{c}(x_0, \rho(o)x_1),$$

where $\tilde{c}(x_0, x_1)$ is given by Equation (16). Hence, this is 200 times more expensive than our approach. This baseline should be much closer to the true batch OT solution. Applying our approximation multiple times reduced the transportation cost slightly. Performing the rotations first, lead to inferior results. We observe the same behavior for the other systems. Therefore, we used the approximation given in Equation (16) throughout the experiments, showing that this is sufficient to learn a close approximation of the OT map.

## A.10 Different data set sizes

Prior work [17, 18] investigates different training set sizes to train equivariant flows, showing that for simple datasets already quite a few samples are sufficient to achieve good NLL. We test if this also holds for flow matching for the more complex alanine dipeptide system. To this end, we run our flow matching experiments also for smaller data set sizes of 10000 and 1000. We report our findings in Table 5. In agreement with prior work, the NLL becomes worse the smaller the dataset. The same is true for the ESS. However, the sampled integration path lengths are nearly independent on the dataset size.

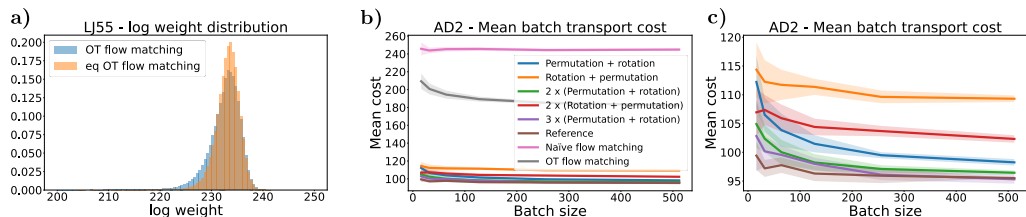

Figure 7: (a) Log weight distribution for samples generated for the LJ55 system. (b, c) Comparison of different approximation methods to approximate Equation (16) for alanine dipeptide.

Table 5: Comparison of flows trained with different training set sizes. Errors over three runs.

| Training type | NLL ($\downarrow$) | ESS ($\uparrow$) | Path length ($\downarrow$) |
|---|---|---|---|
| | Alanine dipeptide - 1000 training samples | | |
| OT flow matching | $\mathbf{-104.97 \pm 0.05}$ | $0.25 \pm 0.04\%$ | $10.14 \pm 0.05$ |
| Eq OT flow matching | $-103.38 \pm 0.03$ | $\mathbf{0.39 \pm 0.08}\%$ | $\mathbf{9.27 \pm 0.02}$ |
| | Alanine dipeptide - 10000 training samples | | |
| OT flow matching | $\mathbf{-105.16 \pm 0.17}$ | $0.33 \pm 0.07\%$ | $10.13 \pm 0.02$ |
| Eq OT flow matching | $-103.36 \pm 0.15$ | $0.40 \pm 0.06\%$ | $\mathbf{9.32 \pm 0.01}$ |

### A.11 Naïve flow matching

We provide naïve flow matching, i.e. flow matching without the OT reordering, as an additional baseline. Naïve flow matching results in even longer integration paths, while the ESS and likelihoods are close to the results of OT flow matching, as shown in Table 6. Flow matching with a non equivariant architecture, e.g. a fully connected neural network, failed for all systems but DW4 and is hence not reported.

## B Proofs and derivations

### B.1 Equivariant OT flow matching

We show that our equivariant OT flow matching converges to the OT solution, loosely following the theory in [67] and extending it to infinite groups. Let $G$ be a compact (topological) group that acts on an Euclidean $n$-space $X$ by isometries, i.e., $d(g \cdot x, g \cdot y) = d(x, y)$, where $d$ is the Euclidean distance. This definition includes the symmetric group $S_n$, the orthogonal group $O(n)$, and all their subgroups. Consider the OT problem of minimizing the cost function $c(x, y) = d(x, y)^p$, $p \geq 1$ between $G$-invariant measures $\nu_1$ and $\nu_2$.

Let us further define the pseudometric $\tilde{d}(x, y)^p = \min_{g \in G} d(x, g \cdot y)^p$. We want to show that any OT map $T : X \to X$ for cost functions $c(x, y) = d(x, y)^p$ is also an OT map for $\tilde{c}(x, y) = \tilde{d}(x, y)^p$.

**Lemma 1.** *Let $\nu_1$ and $\nu_2$ be $G$-invariant measures, then there exists an $G$-invariant OT plan $\pi(g \cdot x, g \cdot y) = \pi(x, y)$.*

Table 6: Results for naïve flow matching. Same hyperparameters as for OT flow matching. Errors over three runs.

| Dataset | NLL ($\downarrow$) | ESS ($\uparrow$) | Path length ($\downarrow$) |
|---|---|---|---|
| DW4 | $1.68 \pm 0.01$ | $93.01 \pm 0.12\%$ | $3.41 \pm 0.02$ |
| LJ13 | $-16.10 \pm 0.01$ | $57.55 \pm 2.20\%$ | $3.77 \pm 0.01$ |
| LJ55 | $-88.44 \pm 0.03$ | $3.15 \pm 1.04\%$ | $8.66 \pm 0.03$ |
| Alanine dipeptide | $-107.56 \pm 0.09$ | $1.42 \pm 0.51\%$ | $11.00 \pm 0.02$ |

*Proof.* Let $\pi' \in \Pi(\nu_1, \nu_2)$ be a (not necessarily invariant) OT plan. We average the OT plan over the group

$$\bar{\pi}(x,y) = \int_G \pi'(g \cdot x, g \cdot y) d\mu(g), \tag{21}$$

where $\mu$ is the Haar measure on $G$.

An important property of the Haar integral is that

$$\int_G f(hg) d\mu(g) = \int_G f(g) d\mu(g) \tag{22}$$

for any measurable function $f$ and $h \in G$. Therefore the average plan is $G$-invariant:

$$\bar{\pi}(h \cdot x, h \cdot y) = \int_G \pi'(hg \cdot x, hg \cdot y) d\mu(g) = \int_G \pi'(g \cdot x, g \cdot y) d\mu(g) = \bar{\pi}(x,y).$$

Next we compute the marginals

$$\int_X \bar{\pi}(x, dy) = \int_X \int_G \pi'(g \cdot x, g \cdot dy) d\mu(g) = \int_G \int_X \pi'(g \cdot x, g \cdot dy) d\mu(g)$$

$$= \int_G \nu_1(g \cdot x) d\mu(g) = \int_G \nu_1(x) d\mu(g) = \nu_1(x)$$

and analogously $\int_X \bar{\pi}(dx, y) = \nu_2(y)$. Hence, the average plan is also a coupling, $\bar{\pi} \in \Pi(\nu_1, \nu_2)$. The cost of the average plan is

$$\int_{X^2} c(x,y) \bar{\pi}(dx, dy)$$

$$= \int_{X^2} c(x,y) \int_G \pi'(g \cdot dx, g \cdot dy) d\mu(g) \qquad \text{(Definition of } \bar{\pi})$$

$$= \int_G \int_{X^2} c(x,y) \pi'(g \cdot dx, g \cdot dy) d\mu(g) \qquad \text{(Fubini's theorem)}$$

$$= \int_G \int_{X^2} c(g^{-1} \cdot x, g^{-1} \cdot y) \pi'(dx, dy) d\mu(g) \qquad \text{(Volume-preserving substitution)}$$

$$= \int_G \int_{X^2} c(x,y) \pi'(dx, dy) d\mu(g) \qquad \text{(Isometric group action)}$$

$$= \int_{X^2} c(x,y) \pi'(dx, dy),$$

and hence the average plan is optimal.

$\square$

**Lemma 2.** *Let $T: x \mapsto y$ be an $G$-equivariant OT map, $x \in X$, and $g^* := \arg\min_{g \in G} c(x, g \cdot T(x))$. Then $g^* = \mathrm{id}$.*

*Proof.* $T$ maps the orbit of $x$, denoted as $G \cdot x = \{g \cdot x | g \in G\}$, to the corresponding orbit of $T(x)$, due to equivariance.

If there exists an $x^* \in X$ such that $g^* \neq \mathrm{id}$ then this property holds true for the entire orbit of $x^*$.

Now, let us define an $G$-equivariant map $T': x \mapsto y$ such that $T'(G \cdot x^*) = g^* T(G \cdot x^*)$ and $T'(x) = T(x)$ everywhere else. However, in this case, $T$ would not be the OT map, as $c(x^*, T'(x^*)) < c(x^*, T(x^*))$. Therefore, it follows that $g^* = \mathrm{id}$ holds for the $G$-equivariant OT map. $\square$

**Discrete optimal transport** Consider a finite number of iid samples from the two $G$-invariant distributions $\nu_1, \nu_2$, denoted as $\{x_i\}_{i=1}^n$, $\{y_i\}_{i=1}^n$, respectively. Let further $\pi(\nu_1, \nu_2)$ be an $G$-invariant OT plan. The discrete optimal transport problem seeks the solution to

$$\gamma^* = \arg\min_{\gamma \in \mathcal{P}} \sum_{ij} \gamma_{ij} M_{ij}, \tag{23}$$

where $M_{ij} = c(x_i, y_j)$ represents the cost matrix and $\mathcal{P}$ is the set of probability matrices defined as $\mathcal{P} = \{\gamma \in (\mathbb{R}^+)^{n \times n} | \gamma \mathbf{1}_n = \mathbf{1}_n / n, \gamma^T \mathbf{1}_n = \mathbf{1}_n / n\}$. The total transportation cost $C$ is given by

$$C = \min_{\gamma \in \mathcal{P}} \sum_{ij} \gamma_{ij} M_{ij}. \tag{24}$$

Approximately generating sample pairs from the OT plan $\pi(\nu_1, \nu_2)$, as required for flow matching (Section 3.4), can be achieved by utilizing the probabilities assigned to sample pairs by $\gamma^*$. Let further

$$\tilde{\gamma}^* = \arg\min_{\gamma \in \mathcal{P}} \sum_{ij} \gamma_{ij} \tilde{M}_{ij} \tag{25}$$

denote the optimal transport matrix for the cost function $\tilde{c}(x, y)$ and $\tilde{C}$ the corresponding transportation cost. Sample pairs $(x_i, y'_j)$ generated according to $\tilde{\gamma}^*$ are oriented along their orbits to have minimal cost, i.e.

$$(x_i, y'_j) = (x_i, g^* y_j), \quad g^* := \arg\min_{g \in G} c(x_i, g \cdot y_j). \tag{26}$$

Note that the group action $g$ does not change the probability under the marginals, as $\nu_1$ and $\nu_2$ are both $G$-invariant.

**Lemma 3.** *Let $\gamma^*$ and $\tilde{\gamma}^*$ be defined as in Equation* (23) *and Equation* (25)*, respectively. Then, sample pairs generated based on $\tilde{\gamma}^*$ have a lower average cost than sample pairs generated according to $\gamma^*$.*

*Proof.* The total transportation cost $\tilde{C}$ is always less than or equal to $C$ since the cost function $\tilde{c}(x_i, y_j) \le c(x_i, y_j)$ for all $i$ and $j$, by definition. Note that as the number of samples approaches infinity, the inequality between $\tilde{C}$ and $C$ becomes an equality. $\square$

Note that in practice, samples generated wrt $\tilde{\gamma}^*$ are a much better approximation, as shown in Section 6. Moreover, by construction, sample pairs generated according to $\tilde{\gamma}^*$ satisfy $\arg\min_{g \in G} c(x_i, g \cdot y_j) = $ id, as indicated in Equation (26). This property, which is also true for samples from the $G$-invariant OT plan according to Lemma 2, ensures that the generated pairs are correctly matched within their respective orbits. However, sample pairs generated according to $\gamma^*$ do not generally possess this property.

We now combine our findings in the following Theorem.

**Theorem 1.** *Let $T : x \mapsto y$ be an OT map between $G$-invariant measures $\nu_1$ and $\nu_2$, using the cost function $c$. Then*

1. *$T$ is $G$-equivariant and the corresponding OT plan $\pi(\nu_1, \nu_2)$ is $G$-invariant.*

2. *For all pairs $(x, T(x))$ and $y \in G \cdot T(x)$ :*

$$c(x, T(x)) = \int_G c(g \cdot x, g \cdot T(x)) d\mu(g) = \min_{g \in G} c(x, g \cdot y) \tag{27}$$

3. *$T$ is also an OT map for the cost function $\tilde{c}$.*

*Proof.* We prove the parts of Theorem 1 in order.

1. Follows from Lemma 1 and the uniqueness of the OT plan for convex cost functions [68].

2. Follows from Lemma 2.

3. Follows directly from 2., Lemma 2, and

$$\int_{X^2} \min_{g \in G} c(x, g \cdot y) \pi(dx, dy) = \int_{X^2} c(x, y) \pi(dx, dy). \tag{28}$$

However, this OT map is not unique.

$\square$

Hence, we can generate better sample pairs, approximating the OT plan $\pi(\nu_1, \nu_2)$, by using $\tilde{c}$ as a cost function instead of $c$. Training pairs $(x_i, y'_j)$ are orientated along their orbits to have minimal cost (Equation (26)).

These findings provide the motivation for our proposed equivariant flow matching objective, as described in Section 4, as well as the utilization of $G$-equivariant normalizing flows, as discussed in Section 5.

### B.2 Mean free update for identical particles

The update, described in Section 5, conserves the geometric center if all particles are of the same type.

*Proof.* The update of layer $l$ does not alter the geometric center / center mass if all particles are of the same type as $m_{ij}^l = m_{ji}^l$ and then

$$
\begin{aligned}
\bar{x}^{l+1} = \sum_i x_i^{l+1} &= \sum_i x_i^l + \sum_{i,j \neq i} \frac{(x_i^l - x_j^l)}{d_{ij} + 1} \phi_d(m_{ij}^l) \\
&= \bar{x}^l + \sum_{i,j > i} \frac{(x_i^l - x_j^l) + (x_j^l - x_i^l)}{d_{ij} + 1} \phi_d(m_{ij}^l) \\
&= \bar{x}^l
\end{aligned}
$$

$\square$

### B.3 Selective reweighting for efficient computation of observables

As normalizing flows are exact likelihood models, we can reweight the push-forward distribution to the target distribution of interest. This allows the unbiased evaluation of expectation values of observables $O$ through importance sampling

$$
\langle O \rangle_\mu = \frac{\mathbb{E}_{x_1 \sim \tilde{p}_1(x_1)}[w(x_1)O(x_1)]}{\mathbb{E}_{x_1 \sim \tilde{p}_1(x_1)}[w(x_1)]}, \quad w(x_1) = \frac{\mu(x_1)}{\tilde{p}_1(x_1)}. \tag{29}
$$

## C  Technical details

### C.1  Code libraries

Flow models and training are implemented in *Pytorch* [69] using the following code libraries: *bgflow* [8, 17], *torchdyn* [70], *Pot: Python optimal transport* [71], and the code corresponding to [18]. The MD simulations are run using *OpenMM*[72], *ASE* [73], and *xtb-python* [56].

The code will be integrated in the bgflow library `https://github.com/noegroup/bgflow`.

### C.2  Benchmark systems

For the DW4 and LJ13 system, we choose the same parameters as in [17, 18].

**DW4**  The energy $U(x)$ for the DW4 system is given by

$$
U^{\mathrm{DW}}(x) = \frac{1}{2\tau} \sum_{i,j} a\,(d_{ij} - d_0) + b\,(d_{ij} - d_0)^2 + c\,(d_{ij} - d_0)^4, \tag{30}
$$

where $d_{ij}$ is the Euclidean distance between particle $i$ and $j$, the parameters are chosen as $a = 0, b = -4, c = 0.9, d_0 = 4$, and $\tau = 1$, which is a dimensionless temperature factor.

Table 7: Model hyperparameters

| Dataset | $L$ | $n_{\text{hidden}}$ | Num. of parameters |
|---------|-----|--------------------|---------------------|
| DW4, LJ13 | 3 | 32 | 22468 |
| LJ55 | 7 | 64 | 204936 |
| alanine dipeptide | 5 | 64 | 147599 |

**LJ13**  The energy $U(x)$ for the LJ13 system is given by

$$U^{\text{LJ}}(x) = \frac{\epsilon}{2\tau} \left[ \sum_{i,j} \left( \left( \frac{r_m}{d_{ij}} \right)^{12} - 2 \left( \frac{r_m}{d_{ij}} \right)^6 \right) \right], \tag{31}$$

with parameters $r_m = 1$, $\epsilon = 1$, and $\tau = 1$.

**LJ55**  We choose the same parameters as for the LJ13 system.

Both the LJ13 and the LJ55 system were generated with MCMC with 1000 parallel chains, where each chain is run for 10000 steps after a long burn-in phase of 200000 steps starting from a random generated initial state.

**Alanine dipeptide**  The alanine dipeptide training data at temperature $T = 300K$ set is generated through two steps: (i) Firstly, we perform an MD simulation, using the classical *Amber ff99SBildn* force-field, at 300K for implicit solvent for a duration of 1 ms [20] using the openMM library.

(ii) Secondly, we relax $10^5$ randomly selected states from the MD simulation for 100 fs each, using the semi-empirical *GFN2-xTB* force-field [56] and the ASE library [73] with a friction constant of 0.5 a.u.

Both simulations use a time step of 1 fs. We create a test set in the same way.

**Alanine dipeptide umbrella sampling**  To accurately estimate the free energy difference, we performed five umbrella sampling simulations along the $\varphi$ dihedral angle using the semi-empirical *GFN2-xTB* force-field. The simulations were carried out at 25 equispaced $\varphi$ angles. For each angle, the simulation was initiated by relaxing the initial state for 2000 steps with a high friction value of 0.5 a.u. Subsequently, molecular dynamics (MD) simulations were performed for a total of $10^6$ steps using the *GFN2-xTB* force field, with a friction constant of 0.02 a.u. The system states were recorded every 1000 steps during the second half of the simulation. A timestep of 1 fs was utilized for the simulations.

The datasets are available at `https://osf.io/srqg7/?view_only=28deeba0845546fb96d1b2f355db0da5`.

## C.3  Hyperparameters

Depending on the dataset, different model sizes were used, as reported in Table 7. All neural networks $\phi_\alpha$ have one hidden layer with $n_{\text{hidden}}$ neurons and *SiLU* activation functions. The embedding $a_i$ is given by a single linear layer with $n_{\text{hidden}}$ neurons.

We report the used training schedules in Table 8. Note that $5e\text{-}4/5e\text{-}5$ in the second row means that the training was started with a learning rate of $5e\text{-}4$ for 200 epochs and then continued with a learning rate of $5e\text{-}5$ for another 200 epochs. All batches were reordered prior to training the model. The only exception is alanine dipeptide trained with equivariant OT flow matching.

## C.4  Parallel OT batch generation

As Equation (16) needs to be evaluated for each possible sample pair in a given batch, the computation cost of equivariant OT flow matching is quite high. However, a simple way to speed up the equivariant OT training is to generate the batches beforehand or in parallel to the training process. This process is highly parallelizable and can be performed on CPUs, which are in practice usually more available

Table 8: Training schedules

| Training type | Batch size | Learning rate | Epochs | Training time |
|---|---|---|---|---|
| | | DW4 | | |
| Likelihood [18] | 256 | 5e-4 | 20 | 3h |
| OT flow matching | 256 | 5e-4/5e-5 | 200/200 | 0.5h |
| Equivariant OT flow matching | 256 | 5e-4/5e-5 | 200/200 | 0.5h |
| | | LJ13 | | |
| Likelihood [18] | 64 | 5e-4 | 5 | 13h |
| OT flow matching | 256 | 5e-4/5e-5 | 1000/1000 | 3h |
| Equivariant OT flow matching | 256 | 5e-4/5e-5 | 1000/1000 | 3h |
| | | LJ55 | | |
| OT flow matching | 256 | 5e-4/5e-5 | 600/400 | 17h |
| Equivariant OT flow matching | 256 | 5e-4/5e-5 | 600/400 | 17h |
| | | Alanine dipeptide | | |
| OT flow matching | 256 | 5e-4/5e-5 | 1000/1000 | 6.5h |
| Equivariant OT flow matching (Batch reordering during training) | 32 | 5e-4/5e-5 | 200/200 | 25h |

Table 9: Wall-clock time to reorder a single batch for the two OT flow matching methods. The times for the DW4 and LJ13 system are below $0.01$ seconds for OT flow matching and are therefore not reported.

| Training type | Batch size | Wall-clock time |
|---|---|---|
| | DW4 | |
| Equivariant OT flow matching | 256 | 3.6s |
| | LJ13 | |
| Equivariant OT flow matching | 256 | 4.5s |
| | LJ55 | |
| OT flow matching | 256 | 0.01s |
| Equivariant OT flow matching | 256 | 20.2s |
| | Alanine dipeptide | |
| OT flow matching | 256 | 0.01s |
| Equivariant OT flow matching | 256 | 22.4s |
| Equivariant OT flow matching | 32 | 0.4s |

than GPUs and also in higher numbers. This also allows for larger batch sizes for the equivariant OT model and comes at little additional cost. Hence, scaling equivariant flow matching to even larger systems should not be an issue. The wall-clock time for the reordering of a single batch on a single CPU is reported in Table 9.

## C.5 Biasing target samples

In the case of alanine dipeptide, the transition between negative and positive $\varphi$ dihedral angles is the slowest process, as depicted in Figure 3d. Since the positive $\varphi$ state is less probable, we can introduce a bias in our training data to achieve a nearly equal density in both states. This approach aids in obtaining a more precise estimation of the free energy. To ensure a smooth density function, we incorporate weights based on the von Mises distribution $f_{\text{vM}}$. The weights $\omega$ are computed along the $\varphi$ dihedral angle as

$$\omega(\varphi) = 150 \cdot f_{\text{vM}}\left(\varphi | \mu = 1, \kappa = 10\right) + 1. \tag{32}$$

We then draw training samples based on the weighted distribution.

## C.6  Effective samples sizes

The effective sample sizes are computed with Kish's equation [74]. We use $5 \times 10^5$ samples for the DW4 and LJ13 system, $2 \times 10^5$ for alanine dipeptide, and $1 \times 10^5$ for the LJ55 system per model.

## C.7  Error bars

Error bars in all plots are given by one standard deviation, averaged over 3 runs, if not otherwise indicated. The same applies for all errors reported in the tables. Exceptions are the *Mean batch transport cost* plots, where we average over 10 batches and the *integration error plots*, where we average over 100 samples for each number of steps.

## C.8  Computing infrastructure

All experiments for the DW4 and LJ13 system were conducted on a *GeForce GTX 1080 Ti* with 12 GB RAM. The training for alanine dipeptide and the LJ55 system were conducted on a *GeForce RTX 3090* with 24 GB RAM. Inference was performed on *NVIDIA A100 GPUs* with 80GB RAM for alanine dipeptide and the LJ55 system.

