# OpenReview forum: "Equivariant flow matching"
_NeurIPS.cc/2023/Conference — NeurIPS 2023 poster_

### Official Review · Reviewer_nzPr · 2023-06-20

**Soundness:** 4 excellent
**Presentation:** 3 good
**Contribution:** 3 good
**Rating:** 6
**Confidence:** 4

**Summary:**

The manuscript builds on recent progress in simulation-free loss functions for continuous normalizing flows which allow to scale CNFs to significantly larger dimensions and can be thought of as a generalizion of continuous time diffusion models. Specifically, the manuscript extends the recently proposed conditional flow matching approach which proposed to use results of optimal transport to construct particular efficient probability paths.

A notable novelty of the paper is to consider the question of how to incorporate equivariance in the conditional flow matching loss. This is achieved by modifying the cost matrix c(x, x') of the Wasserstein distance to be the minimal cost *over the entire orbit* of x' (or equivalently x).

The proposal is studied experimental in the context of Boltzmann generators which are normalizing flows trained to act as sampling density for importance weighting (or, alternatively, Markov Chain Monte Carlo) of unnormalized physical target densities of Quantum Chemistry. Normalizing flows are particularly suited for this task as they provide a tractable likelihood as well as fast sampling. Equivariance is of pivotal importance for Boltzmann generators since the studied physical systems often have a high degree of symmetry, e.g. SE(3) and permutation symmetry.

The paper establishes in detailed numerical experiments that the flow matching approach is beneficial in the context of Boltzmann generators and compares favorably to likelihood-based training. This is shown for standard benchmarks such as Lennard Jones particles and Alanine dipeptide.

Overall, I think however this is a valuable contribution in a very exciting and rapidly evolving field of research. I therefore tend to recommend acceptance. I would however encourage the authors to add an appendix summarising the conditional OT flow matching procedure of 2302.00482 on which their method builds. Unfortunately, the conditional flow matching paper is not the most readable and it would make the ms for self-contained and accessible to add a brief summary.



**Strengths:**

- The paper establishes, to the best of my knowledge, for the first time that flow matching is benefical in the context of Botzmann generators. It also presents a detailed comparison to both likelihood-based as well as OT conditional flow matching training objectives on standard benchmark sets.
- The question of how equivariance is of high relevance as CNFs are widely deployed in physics applications for which symmetry is a fundamental ingredient for successful learning.
- The presentation is well structured and clear


**Weaknesses:**

- The proposed method is rather specific to the case of SE(3) and permutation invariance. It seems non-trival to me how one would extend the treatment to larger symmetry groups, such as in applications to Lattice Field Theories.
- A crucial element of Boltzmann generators is that they are often trained using self-sampled energy training (Variational Inference). The flow matching condition does not allow for such type of training. In the original conditional flow matching paper, a reweighting procedure was proposed. I think the authors made a good choice in not discussing this in this contexts as it is to be expected that reweighting will fail for reasonably sized systems such as alanine dipeptide. Nevertheless, the fact that (an efficient version of) energy-based training is not available for flow matching is a major downside of flow-matching.
- I am a bit unclear on how much the proposed equivariant method actually helps. There is a substantial gain in the LJ55 setup while on Alanine dipeptide the previously proposed OT flow matching leads to higher ESS (at the cost of slightly longer trajectory length).

Minor comments:

P.3 L.80-81: Continuous time diffusion models provide a tractable likelihood. In fact, they are continuous normalizing flows albeit trained with a different objective as can be seen by noticing that the reverse process is equivalent to a deterministic ODE (known as probability flow ODE).

P.3. L.100: distribution -> density

P.3 Eq 3: I find the notation a bit contrived. Why not simply use f_\theta(t, x)?

P.4 L .111: Jacobian trace -> trace of Jacobian or divergence

P.4 Eq 6: x_1 \sim \mu(x_1) -> x_1\sim \mu

P.4 L. 137 \sim q(x_0) -> \sim q and \sim \mu(x_1) -> \sim \mu

P.6 Table 1: Please mention in the caption how the errors are determined. Most readers will wonder about this while looking at the table so it would be good to provide this information already here.

P.6 Eq 15-18: Various commas are missing after the equations

P.6 L. 208: mean free Gaussian is not equal to \mathcal{N}(x_0| 0, 1), as it has different normalizer.


**Questions:**

- How is the minimum over the SO(D) group orbit in Eq (14) actually performed (the group is continuous)?
- What is the advantage of cartesian coordinates for alanine dipeptide as opposed to the standard parameterization in terms of dihedrals, bond angles, ...?
- What is the precise definition of path length in Table 1?

**Limitations:**

Limitations are properly addressed.

---

> ### Author Rebuttal · Authors · 2023-08-10
>
> We thank the reviewer for their insightful review, and appreciate their assessment that our paper is a valuable contribution in a very exciting and rapidly evolving field. We now address their comments individually.
>
> > Unfortunately, the conditional flow matching paper is not the most readable and it would make the ms for self-contained and accessible to add a brief summary.
>
> We agree on the readability of the original flow matching paper. We will include a more detailed discussion both in Sec. 3.4 and in the appendix. We agree that this will make our work more cohere.
>
> > The proposed method is rather specific to the case of SE(3) and permutation invariance. It seems non-trival to me how one would extend the treatment to larger symmetry groups, such as in applications to Lattice Field Theories
>
> We discuss equivariant OT flow matching for more general symmetry groups in appendix B.1 in great detail. We will include the main findings in Sec. 4 in the main part of the manuscript. However, how Eq. 14 can be efficiently approximated for other symmetry groups remains for future research. The $SO(3)$ and $S(N)$ symmetry groups (as well as the translation) are the most important ones for molecular systems, which our paper aims to tackle.
>
> > A crucial element of Boltzmann generators is that they are often trained using self-sampled energy training (Variational Inference). The flow matching condition does not allow for such type of training. In the original conditional flow matching paper, a reweighting procedure was proposed. I think the authors made a good choice in not discussing this in this contexts as it is to be expected that reweighting will fail for reasonably sized systems such as alanine dipeptide. Nevertheless, the fact that (an efficient version of) energy-based training is not available for flow matching is a major downside of flow-matching.
>
> We agree that this is limitation of flow matching. However, it is not feasible to train CNF energy based, as shown in appendix A.2 for the larger systems we investigated. As the reviewer mentions, the proposed reweighting procedure in the flow matching paper is not feasible for larger systems. As the importance weights will be zero, when reweighting from a Gaussian to the target distribution. An alternative approach is to first train a CNF with equivariant OT flow matching, with only a few (potentially biased) samples and then generating new samples with the flow. As these can then be reweighted with non-zero weights, they can then be added to the training set. This could be done iteratively and the final training set would resemble samples from the target equilibrium distribution. We will include this idea in the future work section, as it is beyond the scope of this paper.
>
> > I am a bit unclear on how much the proposed equivariant method actually helps. There is a substantial gain in the LJ55 setup while on Alanine dipeptide the previously proposed OT flow matching leads to higher ESS (at the cost of slightly longer trajectory length).
>
> The difference for equivariant OT compared to normal OT is only marginal for systems with few symmetries, i.e. less than 15 interchangeable particles, which is the case for DW4, LJ13 and ALA2. However, for the much larger LJ55 system (at least 3 times as many identical particles) the difference is significant. Crucially, the straight OT sampling paths allow using a Runge-Kutta integrator instead of the adaptive dopri5, which is significantly faster. With the “normal” OT the particle integration paths change directions (see Figure 2b,c,e), which requires small step sizes where the particles turn. Hence, this shows that it is important to use equivariant OT flow matching when scaling to larger systems to obtain optimal paths. This effect is not that prominent for the smaller systems, where normal OT flow matching already works quite well. We included further experiments in the global rebuttal that highlight the benefits of equivaraint OT flow matching.
>
> > How is the minimum over the SO(D) group orbit in Eq (14) actually performed (the group is continuous)?
>
> It is performed with the Kabsch algorithm, we will include this in the final version of the manuscript. For more details see also the global rebuttal.
>
> > What is the advantage of cartesian coordinates for alanine dipeptide as opposed to the standard parameterization in terms of dihedrals, bond angles, ...?
>
> There are multiple advantages of using Cartesian coordinates over internal coordinates. The main two are transferability and the long *robot arm problem*. (i) Transferability: Training a transferable Boltzmann generator will be easier in Cartesian coordinates, as these are not specific to each system.
> (ii) The long robot arm problem: Peptides and proteins often have different metastable states, which can for example be folded or unfolded. In these folded states some non-bonded parts will be close, but the information that these are close needs to be propagated though the whole backbone structure in the form of torsion angles, angles and bond lengths (long robot arm). Hence, a model in internal coordinates, as it does have the information of this distance implicitly, will fail to learn the distribution accurately. Moreover, going to explicit solvent systems, i.e. with water molecules, internal coordinates will not be possible and accounting for the large amount of possible permutation of interchangeable water molecules will be crucial. Highlighting again the importance of using our equivariant OT flow matching when scaling to larger systems, as shown in the LJ55 results.
>
> > What is the precise definition of path length in Table 1?
>
> The path length as reported in the table and shown in the figures is the usual arc length between two points in $N \times D$ dimensions. We will include this in the final version of the manuscript.
>
> We thank the reviewer for their helpful minor comments and agree with all of them. We will change the final version accordingly.

---

> > ### Comment · Reviewer_nzPr · 2023-08-14
> > **Response to Rebuttal**
> >
> > Thank you for the detailed rebuttal. Your reply clarified my questions. I only have a minor follow-up question:
> >
> > I am confused about the 'transferability' aspect in your last reply. Why is the standard parameterization specific to the individual peptide? As far as I can see any peptide can be expressed using such a representation?

---

> > > ### Author Response · Authors · 2023-08-15
> > >
> > > We appreciate that the reviewer is satisfied with our rebuttal and that we could answer their questions.
> > >
> > > > I am confused about the 'transferability' aspect in your last reply. Why is the standard parameterization specific to the individual peptide? As far as I can see any peptide can be expressed using such a representation?
> > >
> > > While each peptide can indeed be represented in internal coordinates, their descriptions differ significantly due to varying atom counts, leading to distinct numbers and types of torsion angles, angles, and bonds. Inferring from one learned set to another isn't straightforward, especially considering the challenge of constructing a model capable of handling varying internal coordinate counts. In contrast, Cartesian coordinates easily allow varying input sizes. For instance, our approach could be directly applied on different molecules or particle counts.

---

### Official Review · Reviewer_coc5 · 2023-06-27

**Soundness:** 2 fair
**Presentation:** 2 fair
**Contribution:** 2 fair
**Rating:** 3
**Confidence:** 5

**Summary:**

This paper studies equivariant flow matching, an extension to the flow matching paradigm for training continuous normalizing flows by regressing parametric vector fields to conditional vector fields. Building upon prior work, conditional vector fields can be derived from the distribution of the 2-Wasserstein optimal transport map $\pi(x0, x1)$  between the prior $q(x0)$ and the target $\mu(x1)$. The central contribution of this paper is to bake symmetries into the optimal transport cost $c(x_0, x_1)$ through a sequential search procedure for the symmetry group $S(N)$ and $SO(N)$. This is done by first applying the Hungarian algorithm followed by finding a rotation matrix that minimizes the cost, for each element in the cost matrix. Finally, the authors parameterize an equivariant vector field using the EGNN architecture of Satorras et. al 2021. Experiments are done on $n$-body particle dynamics and training a Boltzmann Generator for small proteins in Alanine dipeptide.

**Strengths:**

**Originality**

The main strength of this paper is that all presented material follows naturally from the desire to bake in equivariance in continuous normalizing flows. As equivariance has already been studied extensively in the generative modeling literature this application to flow matching is a reasonable extension. The originality of this work is limited to solely baking symmetries in the cost matrix as equivariant vector fields using EGNN have already been employed in the literature e.g. $E(N)$-normalizing flow (Satorras et. al 2022).

**Quality**

The quality of the presented work is a good first attempt at tackling the problem of equivariant flow matching, but unfortunately, it is below the standard that is expected on several fronts which are outlined in the weaknesses section.

**Clarity**
In general, the work is fairly clear. The presented ideas are straightforward to grasp but a few technical details are omitted which could improve understanding and readability. For example, how do you align the rotations after performing the Hungarian algorithm? We are minimizing over $SO(D)$, this is a non-trivial manifold. Of course, the appendix and code have this information but given that this is a crucial part of the contribution more detail would improve readability.

**Signifance**
As equivariance in generative modeling has largely been studied in the literature the contribution of this work is limited. While equivariance has not been exclusively studied in flow matching, this extension is an early step and has limited novelty. This would be fine if there were large benefits empirically, theoretically, or computationally but this is not the case as far as I can tell and as a result, this paper has limited significance at present.


**Weaknesses:**

This paper has many potential weaknesses some of which are already alluded to in the previous section.
Firstly, it has very little novelty as many of the concepts regarding equivariance and flows are known in the literature. In fact, using EGNN in normalizing flows for this very symmetry group has already been studied by Satorras et. al 2022. Moreover, one of the main benefits of flow matching is that one sidesteps the training complexity of regular CNFs as we do not need to backprop through an ODE solver. This brings huge computational benefits during training, albeit inference is still costly. In this paper, training is also expensive as solving for the optimal coupling in mini-batch OT is already expensive but even more so because the Hungarian algorithm is employed which scales $O(N^3)$. This is prohibitively expensive for any large-scale machine learning system. I encourage the authors to investigate other means of approximately solving for permutations. Some examples and directions for investigation can include learning the permutations (see Git Rebasin Ainsworth et. al 2022) using the straight-through estimator, or using Gumbel Sinkhorn (Mena 2018).

With regard to the proposed approach, the authors break down the problem by doing a sequential minimization by first finding the best permutation and then finding the best rotation. But this is not the original problem which may be indeed intractable. This is a ripe avenue to do a bit of theory to justify the proposed approach. For example, why is this sensible and not the first thing one can do? Can we do a little bit of error analysis to bound the error of the optimal cost matrix found in the sequential search to the actual one?

The experiments section is also rudimentary. While I appreciated the visualization of the shorter and straighter OT-Paths it is difficult to rationalize the gains here knowing that the overall algorithm is likely more expensive. A detailed time-complexity analysis of the overall method is needed here and proper comparison to regular flow matching and $E(N)$ NF is also a good idea. Moreover, the results seem a bit mixed as equivariant models do worse on Alanine dipeptide compared to non-equivariant models. The current explanation in the main text for this result is unsatisfactory. We expect equivariance to help here, why is it not?


**Questions:**


1. One key benefit of equivariant models is that they are more data efficient. This avenue seems to be underexplored. A table similar to Table 1 in $E(N)$-NF would be nice to see.

2. There are many equivariant models for molecular data and simulation but only a small dataset in Alanine dipeptide was used. Can the authors justify not including a large body of equivariant generative models for molecular simulation here?

3. While this paper claims equivariant flow matching as a general procedure the main method is limited to the groups $S(N)$ and $SO(D)$. This is a bit misleading as the current approach for finding the optimal cost matrix is not applicable to other groups. The authors may consider a different title for this work as well as a fairer presentation in the abstract and introduction of what is actually done.

**Limitations:**

Yes

---

> ### Author Rebuttal · Authors · 2023-08-10
>
> We thank the reviewer for their detailed review and questions. To keep the rebuttal within the character limit, we often only cite parts of each question, but always answer the whole one.
>
> > How do you align the rotations after performing the Hungarian algorithm?
>
> With the Kabsch algorithm, see also global rebuttal.
>
> > While equivariance has not been exclusively studied in flow matching, this extension is an early step and has limited novelty. ... this paper has limited significance at present ...
>
> We respectfully disagree regarding the limited novelty and significance. Enhancing MD simulations using ML models is a rapidly growing field, as evidenced by the increasing number of AI4Science groups focused on this topic. Prior to our work, it was uncertain whether training a Boltzmann Generator in Cartesian coordinates to generate samples from the equilibrium distribution of real molecules was possible. We are the first to present a model and training algorithm that successfully achieve this. We believe our results are of great interest to the community and will stimulate further research in this area, potentially leading to transferable Boltzmann Generators.
> Moreover, we consider an even more difficult problem as we train the model with biased samples, instead of sample from the equilibrium distribution. We present two training algorithms that achieve this. Both including the known concepts of flow matching and EGNNs. However, we are the first to combine these, resulting in this novel finding. Additionally, we show that when scaling to larger systems with more symmetries, it becomes critical to do equivariant OT flow matching, to preserve the optimal transport sampling paths. Interestingly our work shows that for systems with less symmetries, e.g. fewer identical particles like DW4, LJ13, and alanine dipeptide, the benefits of including equivariant OT flow matching are marginal (see also appendix A.1). We will highlight these findings more in the final version of the manuscript.
>
> > ... training is also expensive as solving for the optimal coupling in mini-batch OT is already expensive ...
>
> We agree that especially the reordering introduces additional computational cost, as shown in appendix C.3 and discussed in Sec. 8. We share the suggested approximations of the Hungarian algorithm, as detailed in Sec. 8.
> However, the training data preparation can be performed prior to or during the training and is trivially parallelizable on CPUs. Allowing to essentially train the equivariant flow matching models as fast as the OT flow matching models. See the global rebuttal for more details.
>
> > the authors break down the problem by doing a sequential minimization by first finding the best permutation and then finding the best rotation...
>
> We discuss equivariant OT flow matching for more general symmetry groups in appendix B.1 in great detail. We will include the main findings in Sec. 4 in the final version.
> We agree that this minimization problem is intractable, but we show that our approach is close to the correct solution. See global rebuttal for a detailed analysis.
>
> > The experiments section is also rudimentary...
>
> We respectfully disagree that the experiments section is rudimentary. As mentioned, we are the first to train a Boltzmann generator for real molecules in Cartesian coordinates. Nevertheless, we added more experiments and additional evaluations, as shown in the global rebuttal.
> We report runtimes of the training in appendix C.3. However, note that in practice we can do the data preparation in parallel before or during the training.
> We already compare to $E(N)$-NF flows, also on their biased test set in appendix A.3. Note that we do not compare for LJ55 and alanine dipeptide, as likelihood training requires too much memory, as shown in appendix A.2.
> We added comparison to normal flow matching as suggested.
>
> > the results seem a bit mixed as equivariant models do worse on Alanine dipeptide compared to non-equivariant models...
>
> The difference for equivariant OT compared to normal OT is marginal for systems with few symmetries, i.e. less than 15 interchangeable particles, which is the case for DW4, LJ13 and alanine dipeptide. For alanine dipeptide, there are moreover different particle types. However, for the much larger LJ55 system (at least 3 times as many identical particles) the difference is significant. Crucially, the straight OT sampling paths allow using a Runge-Kutta integrator instead of the adaptive dopri5, which is significantly faster. With the “normal” OT the particle integration paths change directions (see Figure 2b,c,e), which requires small step sizes where the particles turn. Hence, this shows that it is important to use equivariant OT flow matching when scaling to larger systems to obtain optimal paths.
>
> > One key benefit of equivariant models is that they are more data efficient. This avenue seems to be underexplored...
>
> We thank the reviewer for this suggestion, we included this experiment in the global rebuttal.
>
> > There are many equivariant models for molecular data and simulation...
>
> We investigate standard benchmarking systems in the literature and even propose new ones, which were previously out of reach for Boltzmann generators in Cartesian coordinates. We leave it for future work to scale to even larger systems. However, we showed that in order to get OT integration paths, equivariant OT flow matching is required when scaling to larger systems.
>
> > While this paper claims equivariant flow matching as a general procedure the main method is limited to the groups $S(N)$ and  $SO(D)$...
>
> As mentioned, the theory for more general symmetry groups is somewhat hidden in appendix B.1. However, the challenge remains of finding a good approximation to eq. 14 for other symmetry groups. Nevertheless, the investigated symmetry groups in our paper are the most important for molecular data. We will change the abstract as suggested.

---

> > ### Comment · Reviewer_coc5 · 2023-08-13
> > **Re:Rebuttal**
> >
> > I thank the authors for their detailed rebuttal. The additional experiments in the global response are appreciated.
> >
> > However, after another careful reading of this work, I will have to, unfortunately, maintain my current position on these papers. The authors claim that the of training Boltzmann generators in Cartesian coordinates using Flow matching is novel and important for AI4Science applications. I disagree on the novelty but I do agree on the potential impact for AI4Science.
> >
> > First, looking at the code provided you require an MCMC step to generate a dataset for NLL-based training. The training part can really be done with any generative model because we have the dataset. The choice of using Flow matching here is, respectfully, not that crazy. Indeed, if you really had to do training and generate samples using ONLY the energy function and no MCMC using something like the reverse KL that would be very intriguing. Unfortunately, this is already done in [1] who also do flow matching.
> >
> > Regarding the experiments in the global response, I seem to have missed where the experiments in the low data regime are done as asked in my original review. I do see Table 4 but this seems to be varying batch size and not dataset size.
> >
> > Regarding the difficulty of the OT problem. I believe the authors will find more interesting ways if they modify the ground cost function $c$, by looking into manifold OT. For Lie groups one idea is to use the geodesic cost instead and that way you do not need to do this approximation. For $S_n$, I do not have a better answer yet.
> >
> > [1] Máté, Bálint, and François Fleuret. "Learning Interpolations between Boltzmann Densities." Transactions on Machine Learning Research (2023).

---

> > > ### Author Response · Authors · 2023-08-14
> > > **Answer to response - Part 1**
> > >
> > > We appreciate the reviewer's extra time spent to reviewing our paper. We now answer their additional responses individually.
> > >
> > > > The authors claim that the of training Boltzmann generators in Cartesian coordinates using Flow matching is novel and important for AI4Science applications. I disagree on the novelty but I do agree on the potential impact for AI4Science.
> > >
> > > We are pleased that the reviewer now acknowledges the potential impact of our work in the AI4Science community. On their assessment about novelty, we respectfully disagree and have elaborated on our stance earlier and below.
> > >
> > > > First, looking at the code provided you require an MCMC step to generate a dataset for NLL-based training.
> > >
> > > We agree that we require data to train with flow matching, which is generally true for flow matching. We mention this in Sec. 3, 6, and in appendix C.2, C.4. However, we do not require training samples from the target distribution, as discussed below.
> > >
> > > > The training part can really be done with any generative model because we have the dataset.
> > >
> > > We respectfully disagree, as we require an exact likelihood model for reweighting to the equilibrium distribution, which differs from the reviewer's assertion that any generative model can be used. Previous research has shown that flows are the most promising approach for addressing this challenge (see Sec. 1 + 2). For the more complex Cartesian coordinate representations, they relied on CNFs due to their enhanced expressiveness. However, applying CNFs to larger systems like alanine dipeptide or LJ55 proved prohibitively expensive, as outlined in appendix A.2. This motivated or solution.
> > >
> > > > The choice of using Flow matching here is, respectfully, not that crazy. Indeed, if you really had to do training and generate samples using ONLY the energy function and no MCMC using something like the reverse KL that would be very intriguing. Unfortunately, this is already done in [1] who also do flow matching.
> > >
> > > We agree that training equivariant flows with flow matching was in proximity, but it remained unexplored until our work. However, as mentioned in the rebuttal, we show that when scaling to larger systems with more symmetries, it becomes critical to do equivariant OT flow matching to preserve the optimal transport sampling paths. Or algorithm for this was not as straightforwardly apparent.
> > >
> > > Respectfully, we differ on the evaluation of [1]. Their approach, evident from Eq 16, does not involve flow matching or simulation-free training. Instead, they introduce a better way of doing energy-based training. Consequently, it is impractical for the larger systems we explored (see appendix A.2). [1] solely assesses their method on small toy systems, raising doubts about its suitability for our larger, more rigid systems, given that one could somehow scale their version of energy based training. Lastly, interpolating the potential as in [1] will generally not lead to OT integration paths.
> > >
> > > We agree with the reviewer's point, that aiming for simulation-free, energy-based training similar to  flow matching holds promise. However, its viability is currently unclear. A potential alternative approach is to initially train a CNF using flow matching with a small set of samples. Subsequent sample generation through the CNF, followed by reweighting to the target distribution, allows these samples to be added iteratively to the training set. Importantly, this process does not rely on backpropagation, enabling scalability for larger systems. While beyond this paper's scope, we will mention this surrogate concept for energy-based training in our future work section.
> > >
> > > Moreover, our approach does not require the training set to originate from the target distribution. This is exemplified in our alanine dipeptide experiments. There, we initially generate samples employing a classical force field and subsequently relax them concerning a three orders of magnitude more expensive semi-empirical force field. As a result, the training samples are bias but are generated significantly faster than solely simulating using the semi-empirical force field. Despite the flow learning from these biased samples, we are able to reweight to the unbiased distribution (see Sec. 6.3). Therefore, this aligns with the reviewer's request, as our method entails notably fewer energy evaluations compared to MD simulations of the target potential, akin to energy-based training. We will clarify this in the final version.

---

> > > > ### Author Response · Authors · 2023-08-14
> > > > **Answer to response - Part 2**
> > > >
> > > > > I seem to have missed where the experiments in the low data regime are done as asked in my original review.
> > > >
> > > > We reported the requested experiments in the low data regime for both the LJ55 and alanine dipeptide system in Table 2 in the global response. Table 1 introduces results for the additional baseline of naïve flow matching, Table 3 reports results for different batch sizes, and Table 4 reports the free energy difference for alanine dipeptide.
> > > > Could the reviewer kindly clarify whether the pdf they reviewed is genuinely distinct or if they might have inadvertently overlooked the results we presented?
> > > >
> > > > >  I believe the authors will find more interesting ways if they modify the ground cost function, by looking into manifold OT. For Lie groups one idea is to use the geodesic cost instead and that way you do not need to do this approximation. For $S_n$, I do not have a better answer yet.
> > > >
> > > > As the reviewer noted, it is unclear how to do manifold OT for S(N), which is the more important and difficult symmetry group, because data augmentation will be infeasible for higher numbers of interchangeable particles. This concern aligns with prior research emphasizing the importance of incorporating permutation symmetry into flows, while using data augmentation for rotations.
> > > > We will include the idea to use manifold OT for SO(3) and combine it with our equivariant flow matching for permutations in the future work section.

---

> > > > > ### Comment · Reviewer_coc5 · 2023-08-17
> > > > > **Re:re Response**
> > > > >
> > > > > I thank the authors for their responses to my questions again.
> > > > >
> > > > > I apologize for missing the results in Table 2. They are indeed the experiments I asked for in the low data regime. The LJ55 experimental findings in table 2 match intuitions and are great. However, the Alanine dipeptide results still bother me. It seems that at 10000 samples Equivariant flow matching does no better than at 1000 samples. Furthermore, both results are significantly worse than the non-equivariant baseline in terms of NLL. This question was also not directly answered in my original review. I would like a definitive answer and a deeper investigation into this matter.
> > > > >
> > > > > > We respectfully disagree, as we require an exact likelihood model for reweighting to the equilibrium distribution, which differs from the reviewer's assertion that any generative model can be used.
> > > > >
> > > > > Disagree. You can train a diffusion model and then use the marginals induced by its probability flow ODE to get exact likelihoods. I am sure the authors know this because they mention it in their response to Reviewer nDvW. I agree that you cannot train say a GAN in this manner but a large class of models (not just flow matching) is permitted.
> > > > >
> > > > > **Faster Training compared to Energy Based Training**
> > > > >
> > > > > (This comment does not apply to Alanine dipeptide experiments) Regarding the faster training for flow matching compared to works in [1]. Of course flow matching using the NLL will be faster than energy-based training. This is because you already have a dataset that is sampled using MCMC. I think the authors will agree that MCMC is extremely expensive for high-dimensional problems, including molecules. Thus you trade off faster training for a big pre-processing step. Note for the biased training for empirical force field where you re-weight using the exact density does not fit into this criticism.

---

> > > > > > ### Author Response · Authors · 2023-08-18
> > > > > > **Re:Re:re Response**
> > > > > >
> > > > > > > I apologize for missing the results in Table 2... the Alanine dipeptide results still bother me. It seems that at 10000 samples Equivariant flow matching does no better than at 1000 samples. Furthermore, both results are significantly worse than the non-equivariant baseline in terms of NLL...
> > > > > >
> > > > > > No worries. We agree that the results for alanine dipeptide are slightly inferiour compared to OT flow matching, even in the low data regime. Note, that the difference lies solely in the training approach, whereas the underlying equivariant architecture remains the same. The results for the OT flow matching technique do not show an improvement with 10000 samples compared to 1000 for alanine dipeptide. This issue does not only appear for equivariant OT flow matching. Hence, achieving lower NLLs appears to necessitate a notably larger dataset. As discussed in our rebuttal, the proposed equivariant OT flow matching performers best when applied to systems possessing greater symmetries. In cases with fewer symmetries, the benefits might be less pronounced. It is possible that the model finds it easier to learn non OT paths for these systems.
> > > > > >
> > > > > >  >> We respectfully disagree, as we require an exact likelihood model for reweighting to the equilibrium distribution...
> > > > > >
> > > > > > > Disagree. You can train a diffusion model and then use the marginals induced by its probability flow ODE to get exact likelihoods. I am sure the authors know this because they mention it in their response to Reviewer nDvW. I agree that you cannot train say a GAN in this manner but a large class of models (not just flow matching) is permitted.
> > > > > >
> > > > > > We do not understand why the reviewer disagrees with our statement, as they provide only an additional example which was already included. We provided a short explanation in our previous comments regarding our preference for a flow model. We kindly request the reviewer to clarify which other alternative generative models they are referring to and how these models could be employed to generate samples from Boltzmann distributions.
> > > > > >
> > > > > > > (This comment does not apply to Alanine dipeptide experiments) Regarding the faster training for flow matching compared to works in [1]. Of course flow matching using the NLL will be faster than energy-based training. This is because you already have a dataset that is sampled using MCMC. I think the authors will agree that MCMC is extremely expensive for high-dimensional problems, including molecules. Thus you trade off faster training for a big pre-processing step. Note for the biased training for empirical force field where you re-weight using the exact density does not fit into this criticism.
> > > > > >
> > > > > > To address any possible misunderstandings, we briefly explain the different loss functions again:
> > > > > > (i) NLL training: This involves evaluating the likelihood of data samples by pushing them through the flow by integration. This is expensive and does not scale to larger systems (appendix A.2).
> > > > > > (ii) Flow matching: The vector field is evaluated at individual points without requiring integration. The training is simulation-free and thus scalable to larger systems.
> > > > > > (iii) Energy-based training: Prior samples are pushed through the entire flow via integration, followed by energy evaluation for the final samples. This is essentially the opposite direction of NLL training and does not scale to the larger systems either. Notably, [1] present a method resembling energy-based training, which shares the same integration-based limitations for scalability to larger systems.
> > > > > >
> > > > > > The reviewer's understanding of why flow matching surpasses energy-based training in speed requires clarification. Flow matching's advantage lies in its avoidance of vector field integration during training, which is the primary bottleneck for NLL and energy-based training, making them unfeasible for larger systems.
> > > > > >
> > > > > > We agree with the reviewer that the data generation becomes more expensive for larger dimensions, because of the more expensive energy evaluation. This is also true for using more accurate force fields. Hence, for efficient training, minimizing the need for costly energy evaluations is important. Energy-based training could address this concern, though its current form lacks scalability. In response, we propose an alternative strategy for molecular experiments, where we minimize the requirement for expensive energy evaluations during data generation by relaxing samples from a less costly simulation. It is important to note that even if energy-based training were to become scalable for larger dimensions, it would still require a comparable number of energy evaluations as our relaxation method, which is much more parallelizable. Furthermore, the reweighting aspect remains the same for both approaches.
> > > > > >
> > > > > > In summary, our method currently stands as the sole scalable solution for larger systems, demanding only a limited number of costly energy evaluations—significantly fewer than conventional MD simulations for molecular systems.

---

### Official Review · Reviewer_nDvW · 2023-07-05

**Soundness:** 3 good
**Presentation:** 3 good
**Contribution:** 3 good
**Rating:** 6
**Confidence:** 4

**Summary:**

The authors propose equivariant flow matching, which provides a way of incorporating syymetries into flow matching objectives.
Specifically, they propose to replace the squared Euclidian distance cost used in the flow matching objective, with (approximately) the minimum squared Euclidian distance over all possible group actions.
They demonstrate in their experiments that this improves training of equivariant flows - obtaining similar or improved performance than vanilla OT flow matching with shorter paths.


**Strengths:**

 - They identify a clear problem of how symmetries (especially permutation symmetries) cause issues with OT flow matching, resulting a (possibly prohibitively) large batch size required.
 - Their solution is simple and fits the problem well.
 - Their experimental results are consistent with the theory (their CNF has shorter paths).

**Weaknesses:**

 - The performance results are mixed - the flow trained with vanilla OT is sometimes better.
Specifically for alanine dipeptide the flow trained with vanilla OT performs better - thus the Equivariant flow matching method does not seem relevant to one of the headline results "for the first time we obtain a Boltzmann generator with significant sampling efficiency without relying on tailored internal coordinate featurization".
On first read of the abstract the wording makes it seem like this result was obtained due to the authors proposed  method rather than applying the existing flow matching technique to the problem.
 - A CNF trained with a classic score matching loss instead of flow OT would be a good obvious baseline but is not included.

**Questions:**

- How was the effective sample size calculated? (equation, number of samples etc)
- The training time for the Equivariant OT flow matching is significantly longer than the vanilla OT flow matching for LJ55 and alanine dipeptide, even though it is trained for less epochs with a smaller batch size - why is this the case? (e.g. is it due to the extra compute required for the search over rotations/permutations?)
- Could you please provide further description on the rationale behind the hyper-parameters used for each experiment in the appendix?
- Could you please specify the compute used for the MD simulation (e.g. number of target energy evaluations, runtime) in the appendix?

**Limitations:**

The Equivariant flow matching method results in longer training times for LJ55 and alanine dipeptide, but no discussion of why / analysis of this is provided (see Questions).
One of the headline results from the abstract (re alanine dipeptide) is obtained using an existing method (without the new proposed method).

---

> ### Author Rebuttal · Authors · 2023-08-10
>
> We thank the reviewer for their detailed review and questions. We now address their comments individually.
>
> > The performance results are mixed - the flow trained with vanilla OT is sometimes better.
>
> The difference for equivariant OT compared to normal OT is only marginal for systems with few symmetries, i.e. less than 15 interchangeable particles, which is the case for DW4, LJ13 and ALA2. However, for the much larger LJ55 system (at least 3 times as many identical particles) the difference is significant. Crucially, the straight OT sampling paths allow using a Runge-Kutta integrator instead of the adaptive dopri5, which is significantly faster. With the “normal” OT the particle integration paths change directions (see Figure 2b,c,e), which requires small step sizes where the particles turn. Hence, this shows that it is important to use equivariant OT flow matching when scaling to larger systems to obtain optimal paths. This effect is not that prominent for the smaller systems, where normal OT flow matching already works quite well.
>
> > Specifically for alanine dipeptide the flow trained with vanilla OT performs better - thus the Equivariant flow matching method does not seem relevant to one of the headline results "for the first time we obtain a Boltzmann generator with significant sampling efficiency without relying on tailored internal coordinate featurization". On first read of the abstract the wording makes it seem like this result was obtained due to the authors proposed method rather than applying the existing flow matching technique to the problem.
>
> We obtain the described result with both the OT flow matching and the equivariant OT flow matching. We extend Table 2 with results obtained from equivariant OT flow matching, see global rebuttal.
> Enhancing MD simulations using ML models is a rapidly growing field, as evidenced by the increasing number of AI4Science groups focused on this topic. Prior to our work, it was uncertain whether training a Boltzmann Generator in Cartesian coordinates to generate samples from the equilibrium distribution of real molecules was possible. We are the first to present a model and training algorithm that successfully achieve this. We believe our results are of great interest to the community and will stimulate further research in this area, potentially leading to transferable Boltzmann Generators.
> Moreover, we consider an even more difficult problem as we train the model with biased samples, instead of sample from the equilibrium distribution. We present two training algorithms that achieve this. Both including the known concepts of flow matching and EGNNs. However, we are the first to combine these concepts, resulting in this novel finding. Moreover, we show that when scaling to larger systems with more symmetries, e.g. as shown for LJ55 with 55 identical Lennard-Jones particles, it becomes critical to do equivariant OT flow matching, to preserve the optimal transport sampling paths. Interestingly our work shows that for systems with less symmetries, e.g. fewer identical particles like DW4, LJ13, and alanine dipeptide, the benefits of including equivariant OT flow matching are marginal and sometimes even show worse performance as in the case of alainine dipeptides. We will highlight these findings more in the final version of the manuscript and change the abstract accordingly.
>
> > A CNF trained with a classic score matching loss instead of flow OT would be a good obvious baseline but is not included.
>
> We included normal flow matching as baseline, see global rebuttal.
>
> > How was the effective sample size calculated?
>
> Kish's equation. We used 10000 samples for the DW4 and LJ13 system and 100000 for LJ55 and alanine dipeptide. We will add the equation, reference, and values to the appendix.
>
> > The training time for the Equivariant OT flow matching is significantly longer than the vanilla OT flow matching for LJ55 and alanine dipeptide, even though it is trained for less epochs with a smaller batch size - why is this the case?  (e.g. is it due to the extra compute required for the search over rotations/permutations?)
>
> Yes, the reviewer's intuition is correct. The training takes longer for the equivariant flow matching because of the search over rotations and especially permutations. However, the training data preparation can be performed prior to or during the training and is trivially parallelizable on CPUs. Allowing to essentially train the equivariant flow matching models as fast as the OT flow matching models. See the global rebuttal for more details.
>
> > Could you please provide further description on the rationale behind the hyper-parameters used for each experiment in the appendix?
>
> The used hyperparameters are discussed in appendix C.3. We performed mostly manual hyperparameter searches and used larger models for the more challenging systems.
>
> > Could you please specify the compute used for the MD simulation (e.g. number of target energy evaluations, runtime) in the appendix?
>
> We will include more details in the appendix. Many simulation details are already included in appendix C.2.
>
> > The Equivariant flow matching method results in longer training times for LJ55 and alanine dipeptide, but no discussion of why / analysis of this is provided (see Questions).
>
> We add a more detailed discussion in the final version, also exploring alternatives and the parallel way to prepare the training data, as discussed above and in the global rebuttal.
>
> > One of the headline results from the abstract (re alanine dipeptide) is obtained using an existing method (without the new proposed method).
>
> The novel result of training a Boltzmann generator for molecules in Cartesian coordinates is obtained with both the equivariant OT flow matching and OT flow matching. See the discussion above as well as the global rebuttal.

---

> > ### Comment · Reviewer_nDvW · 2023-08-14
> >
> > Thank you for your detailed response. I have the following remaining concern:
> > > The novel result of training a Boltzmann generator for molecules in Cartesian coordinates is obtained with both the equivariant OT flow matching and OT flow matching.
> >
> > It seems that the paper could be split into two key contributions:
> > (1) Applying flow matching to the Boltzmann distribution of molecules
> > and
> > (2) Improving flow matching with equivariant flow matching.
> >
> > Currently I think the abstract strongly emphasizes the contribution (2), and I think the typical reader would conclude from reading the following claim in the abstract
> > > where for the first time we obtain a Boltzmann generator with significant sampling efficiency without relying on tailored internal coordinate featurization
> >
> > is only possible with the Equivariant flow matching technique.
> > However, in their rebuttal the authors are strongly emphasizing contribution (1) - which upon my first reads I took as more of a "baseline" than a contribution.
> > I agree that both of these contributions are valuable.
> > However, I agree with Reviewer coc5 that (1) is not very novel as there is already work that uses score matching for cartesian coordinate molecular data (https://arxiv.org/abs/2203.17003) with equivariant networks which is very similar in problem/solution structure (all though of course there are some differences).
> >
> > I think differentiating contributions (1) and (2) in the abstract would clarify the paper - to make sure the presented claims are accurately interpreted.
> > Lastly, for contribution (1) to be a significant contribution I think the paper would need to provide more insight into some specifics of training CNFs as Boltzmann generators (such as training by energy) and detailed comparisons to existing approaches (such as comparing to discrete flows on internal coordinates).

---

> > > ### Author Response · Authors · 2023-08-15
> > >
> > > We appreciate the reviewer's extra time spent to reviewing our paper. We now answer their additional responses individually, again only citing parts of each question, but answering the whole one.
> > >
> > > > ... I agree that both of these contributions are valuable. However, I agree with Reviewer coc5 that (1) is not very novel as there is already work that uses score matching for cartesian coordinate molecular data (...) with equivariant networks which is very similar in problem/solution structure (all though of course there are some differences).
> > > I think differentiating contributions (1) and (2) in the abstract would clarify the paper - to make sure the presented claims are accurately interpreted.
> > >
> > > We acknowledge that the initial abstract formulation might have caused confusion. As recommended by the reviewer, we will emphasize the two primary contributions throughout the abstract, introduction, and conclusion. We appreciate the reviewer's recognition of the value of both these contributions.
> > >
> > > We concur that the work by Hoogeboom et al. addresses a related issue using a similar method, as we acknowledge in Section 2. However, their focus lies in generating molecular conformers rather than sampling from the Boltzmann distribution, employing a diffusion model instead of a flow. Notably, their conformer generation process does not necessitate reweighting to the target distribution, thus avoiding the utilization of the corresponding probability flow ODE for sampling, which would have been closer to our approach.
> > > Moreover, previous studies on flow matching have demonstrated that score-based models yield significantly longer integration paths compared to even naïve flow matching. Consequently, generating samples with the probability flow ODE becomes more expensive.
> > >
> > > We agree that training equivariant flows with flow matching was in proximity, but it nevertheless remained unexplored until our work.
> > >
> > > > Lastly, for contribution (1) to be a significant contribution I think the paper would need to provide more insight into some specifics of training CNFs as Boltzmann generators (such as training by energy) and detailed comparisons to existing approaches (such as comparing to discrete flows on internal coordinates).
> > >
> > > In our revised version, we will provide a more detailed explanation than in the initial paper to explain why CNFs combined with flow matching offer the most promising results and future potential for generating Boltzmann distribution samples.
> > >
> > > Energy based training is currently impossible with simulation free training. Hence, it is infeasible for CNFs for larger systems, as shown in appendix A.2. A potential alternative approach is to initially train a CNF using flow matching with a small set of samples. Subsequent sample generation through the CNF, followed by reweighting to the target distribution, allows these samples to be added iteratively to the training set. Importantly, this process does not rely on backpropagation, enabling scalability for larger systems.
> > >
> > > Moreover, our approach does not require the training set to originate from the target distribution. This is exemplified in our alanine dipeptide experiments. There, we initially generate samples employing a classical force field and subsequently relax them concerning a three orders of magnitude more expensive semi-empirical force field. As a result, the training samples are biased but are generated significantly faster than solely simulating using the semi-empirical force field. Despite the flow learning from these biased samples, we are able to reweight to the unbiased distribution (Sec. 6.3). This training requires a similar amount of energy evaluations as traditional energy based.
> > >
> > > There are multiple advantages of using Cartesian coordinates over internal coordinates for molecular systems. The main two are (i) transferability and (ii) the long *robot arm problem*: (i) Training a transferable Boltzmann generator will be easier in Cartesian coordinates, as these are not specific to each system. The internal coordinate descriptions differ significantly for varying atom counts.
> > > (ii) For example, in folded states some non-bonded parts will be close, but the information that these are close needs to be propagated though the whole backbone structure in the form of torsion angles, angles and bond lengths (long robot arm). Hence, a model in internal coordinates, as it does have the information of this distance implicitly, will fail to learn the distribution accurately. Moreover, going to explicit solvent systems, i.e. with water molecules, internal coordinates will not be possible and accounting for the large amount of possible permutation of interchangeable water molecules will be crucial. Highlighting again the importance of using our equivariant OT flow matching when scaling to larger systems, as shown in the LJ55 results. Note that it is difficult to represent the Lennard-Jones clusters (LJ13, LJ55) in internal coordinates.

---

> > > > ### Comment · Reviewer_nDvW · 2023-08-15
> > > >
> > > > Thank you for your response. I think the authors have demonstrated that they are on the same page as me with regards to addressing the concerns that I have raised.
> > > > Overall, although I do not think this paper is very novel, I think after adressing the aforementioned issues it will still be a valuable contribution to Boltzmann Generator literature.
> > > > Hence, I am happy to stick to my reccomendation of weak accept and do not have any further questions.

---

### Official Review · Reviewer_W92U · 2023-07-08

**Soundness:** 3 good
**Presentation:** 3 good
**Contribution:** 2 fair
**Rating:** 7
**Confidence:** 4

**Summary:**

This paper extends on existing works on Flow Matching with minibatch OT solutions to the case of invariant cost functions. In particular, where the invariance is given by an SO (permutation + rotation) group. This is mainly a method of correcting the minibatch bias, since (non-equivariant) minibatch OT will still converge to the correct mapping when the minibatch size goes to infinity. Empirically, it is shown that equivariance OT matching results in lower transport costs, and hence shorter path lengths which may indicate that it is computationally faster to simulate after training than non-equivariant OT (though not shown).

**Strengths:**

  - Well-written and easy to understand the high-level idea.

  - Straightforward extension of existing works to training equivariant flows.

**Weaknesses:**

  - Eq 13 and 14 are not easy to solve, and it is unclear how much compute (or wall-clock time) these subproblems require.

  - Lack of comparison to other baselines (equivariant diffusion models, standard flow matching).

  - The empirical differences between equivariant OT and OT seem marginal.

  - While the paper discusses transport cost, it is not shown that these translate to faster sampling algorithms.

**Questions:**

  - Eq 14 gives a suboptimal solution to Eq 13. Has the authors looked into how this affects either the transport cost or the final trained model? For instance, one could perhaps solve in an alternating fashion the optimal permutation and rotation. The current method can be seen as a single step approximation of this procedure.

  - It seems that all of the experiments are on fitting to potential functions. However, the main algorithm assumes data is sampled IID. This mismatch between the problem statement and benchmark problems seems to not be discussed much at all. Is my understanding correct that the training datasets are all simulated from some MCMC procedure?

  - Related to the above, why not experiment on data sets such as QM9, which seems to be used by prior works such as equivariant diffusion models [1]?

  - Table 1 doesn't seem to show large improvement for equivariant OT. Why is that? From what I understand, at small batch sizes, equivariant OT should perform better. But is it perhaps that at regular batch sizes, the difference between OT and equivariant OT becomes much smaller?

  - Also, it would be interesting to see how the regular flow matching performs here, since flow matching with conditional OT paths can approximate OT solutions quite well already [2], and there has been a few papers discussing the relation between standard diffusion models and optimal transport [3].

  - As part of future work section, authors discuss approximations to solving OT problems. It might be worth testing out the approximate algorithms proposed in the published work on minibatch OT + flow matching [2], which the authors should cite. Here they showed heuristic algorithms that have similar performance to minibatch OT but with faster compute cost.

  - Regarding the transport cost, one reason for wanting a low transport cost that this reviewer is familiar with, is the sampling efficiency. However, this aspect isn't shown quantitatively in the experiments. For instance, a plot of ESS (or any performance metric) vs NFE, comparing OT and equivariant OT, would be very useful. From the figures, it is not strongly convincing that equivariant OT is actually performing significantly better than OT yet.

  - Another avenue perhaps is a plot of ESS vs batch size. It would be good to understand at what batch sizes, equivariant OT exhibits better performance than OT.

  - Showing wallclock time of equivariant OT, OT and regular flow matching would be ideal. For instance, Table 9 of [2] shows that using minibatch OT solutions barely increases compute by 4%, but I think having to solve for optimal permutation (especially when the number of particles is high) is a much harder task. It isn't clear whether the increase in compute cost is a good tradeoff for better transport cost (I can believe it is; just that it isn't shown in the paper). For instance, a convergence plot vs wallclock time would be very convincing.

  - As a reviewer who is not familiar with the experimental setups and cannot recognize these systems by name, it would be useful to have a table summarizing each experiment. For instance, list out what is the potential function, and how many particles are in the system? This can help provide a better sense of how challenging each task is.

  - Finally, I am just somewhat perplexed by the motivation of training an equivariant generative model from potential functions, when the training aspect requires MCMC sampling from the desired potential function as a first step. This seems to nullify the simulation-free aspect of flow matching, and it seems that an approach for posterior inference would be much better suited. At the same time, I'm sure the proposed algorithm could work well in settings where a dataset is provided (e.g. QM9 seems to be standard), but these experiments do not appear in this paper.

[1] "Equivariant Diffusion for Molecule Generation in 3D" ICML 2022. https://arxiv.org/abs/2203.17003.

[2] "Multisample Flow Matching: Straightening Flows with Minibatch Couplings". ICML 2023. https://arxiv.org/abs/2304.14772.

[3] "Understanding DDPM Latent Codes Through Optimal Transport". ICLR 2023. https://openreview.net/forum?id=6PIrhAx1j4i.

----

I am satisfied with the authors' answers and have increased my score in light of the new experiments and clarifications. See my reply for some comments regarding the rebuttal.

**Limitations:**

Overall, I think the benefits of equivariant OT compared to OT (and regular flow matching) can be showcased more. Theoretically, I can believe what the authors are claiming, but empirically I don't quite see these emphasized as part of the empirical results yet. I suggested some ideas for additional plots in the Questions section.

---

> ### Author Rebuttal · Authors · 2023-08-10
>
> We thank the reviewer for their detailed review and questions. We now address their comments individually. To keep the rebuttal within the character limit, we often only cite parts of each question.
>
> > Eq 13 and 14 are not easy to solve ...
>
> Eq 13 is indeed intractable, but Eq 14 gives a good tractable approximation, see global rebuttal.
>
> > Lack of comparison to other baselines ...
>
> We added the suggested standard flow matching baseline in the global rebuttal.
>
> > The empirical differences between equivariant OT and OT seem marginal.
>
> The difference for equivariant OT compared to normal OT is only marginal for systems with few symmetries, i.e. less than 15 interchangeable particles, which is the case for DW4, LJ13 and ALA2. However, for the much larger LJ55 system (at least 3 times as many identical particles) the difference is significant. Crucially, the straight OT sampling paths allow using a Runge-Kutta integrator instead of the adaptive dopri5, which is significantly faster. With the “normal” OT the particle integration paths change directions (see Figure 2b,c,e), which requires small step sizes where the particles turn. Hence, this shows that it is important to use equivariant OT flow matching when scaling to larger systems to obtain optimal paths. This effect is not that prominent for the smaller systems, where normal OT flow matching already works quite well. See also the global rebuttal.
>
> > While the paper discusses transport cost, it is not shown that these translate to faster sampling algorithms.
>
> We report in section 6.2 a speed-up of about 10 for the inference for the LJ55 system. Moreover, we performed additional experiments to highlight his further, see global rebuttal.
>
> > Eq 14 gives a suboptimal solution to Eq 13...
>
> For the discussed systems, our approximation is quite close. We also tested the suggested approximation algorithm, see global rebuttal.
>
> > It seems that all of the experiments are on fitting to potential functions...
>
> The algorithm does not require IID samples, the samples can be biased. In fact, we show this for the molecule experiments, where the training samples do not stem from the target potential, which is a semi-empirical potential. Instead, we generate the samples with a classical force field, which is about three orders of magnitude cheaper, and then relax these samples with respect to the semi-empirical force field. Hence, the training samples are biased and not IID. Although, the flow then learns a biased potential function, we can reweight the flow samples to the unbiased distribution (see appendix B.3). This is potentially much faster than doing the simulation wrt to the semi-empirical force field. We further validate that we indeed generate samples from the unbiased distribution by comparing to an umbrella simulation (Sec 6.3).
>
> > why not experiment on data sets such as QM9, ...
>
>  QM9 is a different task, i.e. conformer generation (single conformations instead of the Boltzmann distribution). Moreover, the QM9 dataset consists of only small molecules and the aim of this paper is to scale to Boltzmann Generators for the first time to significantly larger systems.
>
> > Table 1 doesn't seem to show large improvement for equivariant OT ...
>
> See above. We investigated different batch sizes in the global rebuttal.
>
> > ... how the regular flow matching performs here,...
>
> We added this baseline, see global rebuttal.
>
> > ... approximations to solving OT problems ...
>
> We will cite that in the related work section. See global response.
>
> > ... plot of ESS  vs NFE... and a plot of ESS vs batch size ...
>
> These are excellent suggestion. We compare as suggested, highlighting the importance of equivariant flow matching when scaling to larger systems. See general rebuttal.
>
> > ... wallclock time ...
>
> We compare the wall-clock training time in appendix C.3 Table 5. The training does take longer for the equivariant flow matching. However, a simple way to speed up the equivariant OT training is to generate the batch pairs not during training. This process is highly parallelizable and can be performed on CPUs, which are in practice usually more available than GPUs. See global rebuttal for the suggested plot in Figure 1h.
>
> > ... table summarizing each experiment
>
> We agree that the information of the dataset should be more visible. We included most of the information only in the appendix C.2 and will move parts to the main part in the final version of the manuscript.
>
> > ... motivation of training an equivariant generative model from potential functions, when the training aspect requires MCMC sampling from the desired potential function as a first step...
>
> We show for Alanine dipeptide that we do not require samples from the equilibrium target Boltzmann distribution, as discussed above. This is a common setting, as the target potential function is often known up to a constant, but sampling from the equilibrium distribution is difficult. However, generating biased sample, e.g. in different meta-stable states, is often feasible, which can then be used to train a Boltzmann Generator and produce unbiased samples.
> Unfortunately, posterior inference requires backpropagating through the whole integration path, which is infeasible for the larger system, as discussed in appendix A.2. Note that also for posterior inference, we usually require some data from the target distribution for initial training of the model.
> However, our work paves the way for transferable Boltzmann generators, as we operate in Cartesian coordinates. These will allow training on (biased) trajectories of a few molecules and be applicable to unseen ones. Hence, requiring simulations only for the training molecules. We leave this exciting avenue for future research.

---

> > ### Comment · Reviewer_W92U · 2023-08-16
> >
> > I thank the authors for their response and updated evaluations, especially regarding ESS vs NFE and the naive flow matching (but with equivariant architecture) approach. To me, I feel that the technical contribution of equivariant OT extension is quite interesting and sufficiently novel. However, the application of learning equivariant generative models for stationary distributions seems like it is a mere subset of the potential applications an equivariant flow matching model can be used for. I agree this is a difficult problem, I just don't understand the motivation for focusing on this and not other equivariant data distributions.
> >
> > While I am not familiar with the literature around Boltzmann Generators, I do agree with the authors that the work (Máté, Bálint, and François Fleuret. "Learning Interpolations between Boltzmann Densities.") brought up by another reviewer is closer to a physics-informed neural net approach of fitting PDEs, which wouldn't scale. On the other hand, one could also say that this work requires sampling (either exactly or approximately) from the stationary distribution beforehand which adds additional complexity and reliance on the sampling algorithm.
> >
> > Overall though, I appreciate the new plots in the rebuttal (namely, figures b and d) and I am still in favor of accepting this work.

---

> > > ### Author Response · Authors · 2023-08-18
> > >
> > > We thank the reviewer for their additional time spent reviewing our paper. We will now address their additional comments below.
> > >
> > > > I thank the authors for their response and updated evaluations, especially regarding ESS vs NFE and the naive flow matching (but with equivariant architecture) approach. To me, I feel that the technical contribution of equivariant OT extension is quite interesting and sufficiently novel. However, the application of learning equivariant generative models for stationary distributions seems like it is a mere subset of the potential applications an equivariant flow matching model can be used for. I agree this is a difficult problem, I just don't understand the motivation for focusing on this and not other equivariant data distributions.
> > >
> > > We are thankful for the reviewer's acknowledgment of our rebuttal's effectiveness and their positive assessment of the novelty and significance of our equivariant OT flow matching algorithm.
> > >
> > > We agree that applying flow matching to equivariant flow models could theoretically extend to molecular conformer generation. However, it is crucial to recognize that this represents a fundamentally distinct problem, potentially not best suited for the flow matching approach. Unlike the sampling task from the Boltzmann distribution that we address, determining the Boltzmann distribution for individual molecules in datasets like QM9 or GEOM is not feasible nor the learning objective. This eliminates the need for reweighting generated samples to match a target Boltzmann distribution, rendering an exact likelihood model unnecessary.
> > > Furthermore, these datasets often contain only a few samples per molecule. This sparse representation might pose challenges for OT flow matching, as the limited sample count prevents the reordering of batches to generate optimal transport paths. It is due to these reasons that we have chosen to focus on sample generation from Boltzmann distributions rather than pursuing the conformer generation problem.
> > >
> > > > While I am not familiar with the literature around Boltzmann Generators, I do agree with the authors that the work (Máté, Bálint, and François Fleuret. "Learning Interpolations between Boltzmann Densities.") brought up by another reviewer is closer to a physics-informed neural net approach of fitting PDEs, which wouldn't scale. On the other hand, one could also say that this work requires sampling (either exactly or approximately) from the stationary distribution beforehand which adds additional complexity and reliance on the sampling algorithm.
> > >
> > > We agree with the reviewer that the work of Máté et al. does not scale to the larger systems we investigated, as they require integration for their energy based loss function and, hence, can not perform simulation free training. However, any energy based training is currently impossible to do simulation free. Hence, it is infeasible for CNFs for larger systems, as shown in appendix A.2. A potential alternative approach is to initially train a CNF using flow matching with a small set of samples. Subsequent sample generation through the CNF, followed by reweighting to the target distribution, allows these samples to be added iteratively to the training set. Importantly, this process does not rely on backpropagation, enabling scalability for larger systems, contrary to energy based training.
> > >
> > > Moreover, as detailed in our initial rebuttal, our approach does not require the training set to originate from the target distribution. This training method then requires a similar amount of energy evaluations as traditional energy based (or the advanced energy based training as proposed by Máté et al.).

---

### Author Rebuttal · Authors · 2023-08-10

We thank the reviewers for their time reviewing our paper and for their insightful questions and suggestions.
We present results for most of the suggested additional experiments and evaluations below and in the attached pdf. Moreover, we address some common questions and clarifications as well.

**The theory for equivariant flow matching** for more general symmetry groups was somewhat hidden in appendix B.1 We will move the main findings to Sec. 4 and highlight this more in the abstract as suggested.

**Approximation of Eq 13** The approximation given in Eq. 14 is in practice performed using the Hungarian algorithm for permutations and the Kabsch algorithm for rotations. We compare several different other suggested approximations of Eq. 13 in Fig. 1c,d in the pdf. The baseline reference is computed with an expensive search over the approximation given in Eq. 14. Namely, we evaluate Eq. 14 for 100 random rotations combined with the global reflection, denoted as $O_{200}$, for each sample, i.e.
$\hat{c}(x_0, x_1)=\min_{o\in O_{200}(D)}\tilde{c}(x_0, \rho(o) x_1),$ where $\tilde{c}(x_0,  x_1)$ is given by Eq 14.
Hence, this is 200 times more expensive than our approach. This baseline should be much closer to the true batch OT solution.
The presented results for alanine dipeptide show that our approximation is simple, while also being a very good approximation, which we also show in our experiments in the pdf and the main paper. Applying our approximation multiple times reduced the transportation cost slightly. Performing the rotations first, lead to inferior results. We observe the same behavior for the LJ55 system.

**Parallel batch generation** In the current version of the paper, we perform the batch preparation during training.
However, a simple way to speed up the equivariant OT training is to generate the batches beforehand or in parallel to the training process. This process is highly parallelizable and can be performed on CPUs, which are in practice usually more available than GPUs and also in higher numbers. This also allows for larger batch sizes for the equivariant OT model and comes at little additional cost. Hence, scaling equivariant flow matching to even larger systems should not be an issue. We use this procedure for the new experiments and will mention this in the final version.

**Naïve flow matching baseline** We include naïve flow matching with the same equivariant architecture as an additional baseline as requested by multiple reviewers. Naïve flow matching results in even longer integration paths, as shown in Figure 1 and Table 1. The other results are close to the results of OT flow matching.
Flow matching with a non equivariant architecture, i.e. a dense neural network, failed for all systems but DW4 and is hence not reported.

**Benefits of equivariant OT flow matching**
We show that when scaling to larger system sizes, e.g. the LJ55 system, equivariant OT flow matching is crucial to maintain optimal integration paths, which result in significantly faster sampling (Figure 1b) and allows the usage of fixed step integrators. Moreover, also the training is faster for the equivariant OT models, as they converge faster (Figure 1h).

**Training set sizes**
We compare different training set sizes for alanine dipeptide and LJ55 in Table 2 in the pdf. Especially the integration paths lengths are the same for the different training set sizes. Note that we used $10^6$ training samples for all the results in the main paper. All other hyperparameters are the same. As observed in prior work, equivariant models are quite data efficient, which is also reproduced by our findings.

**Batch sizes**
We compare different batch sizes for the LJ55 system in Table 3 in the pdf. The integration paths lengths are again similar across the different batch sizes. However, smaller batch sizes resulted in better likelihoods for both flow matching models.

**Simulation details**
We prepare the training data before training as described above.
We will use more runs for the error calculation and more samples to estimate ESS in the final version.
The reported ESS might be misleading as we do not generate many samples and the negative log likelihoods are higher than observed for the larger training set sizes. Hence, some states might be missed by the model for the LJ55 system.

We will merge Table 1 in the pdf with Table 1 in the paper. Table 2 and 3 in the pdf will be added to the appendix, while the results will be discussed in the main part. Table 4 in the pdf will replace Table 2 in the paper.

We thank the reviewers for their valuable suggestions and hope for an engaging discussion period.

---

### Decision · Program_Chairs · 2023-09-21

**Decision:**

Accept (poster)

**Comment:**

This paper extends Flow Matching to equivariant flows. The paper is well written. It also uses symmetry to reduce generation path length and find better coupling of noise and data for training. The main issues in the paper are: It has somewhat limited baselines, missing time-complexity (e.g., in the context of solving eqs. 13,14), marginal improvement of symmetric-aware sampling over independent coupling of noise and data, missing motivation for the MCMC sampling as a first step to FM training, missing link of transport cost and faster sampling, and unified contribution claims of equivariant FM and better symmetry-induced coupling. During rebuttal the authors agreed to address these issues, and incorporate relevant changes in the camera ready version.